

# Reciprocal bias compensation and ensuing uncertainties in model-based climate projections: pelagic biogeochemistry versus ocean mixing.

Ulrike Löptien[1,2] and Heiner Dietze[1,2]

[1]Institute for Geosciences, University of Kiel, Ludewig-Meyn-Str. 10, 24118 Kiel, Germany
[2]GEOMAR Helmholtz Centre for Ocean Research Kiel, Düsternbrooker Weg 20, 24105 Kiel, Germany

**Correspondence:** U. Löptien (ulrike.loeptien@ifg.uni-kiel.de)

**Abstract.** Anthropogenic emissions of greenhouse gases such as $CO_2$ and $N_2O$ impinge on the earth system which, in turn, modulates atmospheric greenhouse gas concentrations. The underlying feedback mechanisms are complex and, at times, counterintuitive. So-called *Earth System Models* have recently matured to standard tools tailored to assess these feedback mechanisms in a warming world. Respective model applications range from being targeted at basic process understanding to the as-
5  sessment of geo-engineering options. A problem endemic to all these applications is the need to estimate poorly known model parameters, specifically for the biogeochemical component, based on observational data (e.g. nutrient fields). In the present study, we illustrate that by such an approach biases in the physical ocean-circulation model component of an Earth System Model can reciprocally compensate biases in the pelagic biogeochemical model component (and vice versa). We present two configurations of an Earth System Model that share a remarkably similar steady state (based on ad-hoc measures) when driven by historical boundary conditions, even though they feature substantially different configurations (sets) of ocean-mixing and
10  biogeochemical cycling (model parameters). When projected into the future the similarity between the model responses breaks. Metrics like total oceanic carbon content and suboxic volume diverge in the model configurations as the Earth warms. Our results reiterate that advancing the understanding of oceanic mixing processes will reduce the uncertainty of future projections of the oceanic biogeochemical cycles. Vice versa, we suggest that an advanced understanding of oceanic biogeochemical cycles can be used for advancements in the ocean circulation modules.

## 1 Introduction

Challenges associated with global warming trigger a discussion of geo-engineering options. Among these options are ideas to purposely change pelagic biogeochemical cycles in order to increase oceanic carbon sequestration away from the atmosphere (e.g., Williamson et al., 2012). Currently, the effectivity and potential side effects of such measures are quantified with numeri-
20  cal Earth System Models (cf., Yool et al., 2009; Dutreuil et al., 2009; Oschlies et al., 2010)) - tools known to be associated with substantial uncertainty (e.g., Bopp, 2013; Friedlingstein et al., 2014). One source of uncertainty in these models are unknowns in the mathematical representation (typically a set of partial differential equations) of both physical and biogeochemical processes in the pelagic ocean.





As for the biogeochemical processes considerable uncertainty is associated to poorly known model parameters (cf., Kriest et al., 2010; Löptien and Dietze, 2017), such as growth rates or limitation thresholds that govern the dynamics of the set of underlying partial differential equations. In an attempt to reduce this uncertainty, several studies set out to estimate these parameters by minimizing a *cost function*, that measures the misfit between model and observations, such as climatological

nutrient and/or phytoplankton concentrations (e.g., Fan and Lv, 2009; Friedrichs et al., 2006; Schartau, 2003; Spitz et al., 1998; Hemmings and Challenor, 2012; Matear, 1995; Xiao and Friedrichs, 2014)). For typical generic problems (as opposed to idealized special cases) it, unfortunately, turned out to be impossible to determine an optimal parameter set (Ward et al., 2010; Schartau et al., 2001; Rückelt et al., 2010). Among the suggested reasons for such failures are excessive computational expenses along with sparse and noisy observational data (e.g., Lawson et al., 1996; Löptien and Dietze, 2015). In addition to,

or as a consequence of, the discussion of the problem entails suggestions that the optimization problem is underdetermined (Matear, 1995) and that the underlying equations do not represent actual processes and conditions (Fasham et al., 1995; Fennel et al., 2001). In this study we will illustrate the connatural problems that come along with biases and deficiencies in the physical model (cf., Sinha et al., 2010; Dietze and Löptien, 2013).

Typically, general ocean circulation models, designed to simulate the ocean's physics such as the movement and mixing

of water parcels (and therein dissolved or dispersed substances of biogeochemical relevance), are contain various sources for uncertainties. Such uncertainties result, e.g., from discretization issues to unresolved processes at the atmosphere-ocean interface (that supply the energy for ocean currents and turbulence). In the context of pelagic biogeochemical cycles, major uncertainty is associated with energy dissipation and related diapycnal mixing. The reason being that diapycnal (mixing) transport of nutrients to the sun-lit surface ocean fuels autotrophic growth of phytoplankton, thus balancing the associated

vertical (sinking) export of organic carbon to depth, away from the atmosphere. As such diapycnal mixing is key to what is also referred to as the *biological pump of carbon*. However, even though diapycnal mixing is key also in determining various physical properties, such as the simulated thermocline depth (e.g. Bryan, 1987) and the simulated global meridional overturning circulation (e.g. Prange et al., 2003), it is not yet well quantified by observations: large-scale tracer release experiments in the thermocline of the oligotrophic suptropical North Atlantic suggest diffusivities between 0.1 to 0.5 $cm^2/s$ (Ledwell et al., 1998)

while measurements which apply the Osborn-Cox relation between dissipation and diffusion exceed these values locally over rough topography by an order of magnitude.

Somewhat disconcerting, effective diapycnal mixing is not even quantifiable in ocean general circulation models where actual mixing compounds explicitly prescribed mixing rates with spurious mixing triggered by numerical advection and isopycnal diffusion algorithms (Mathieu and Deleersnijder, 1998; Lemarié et al., 2012). Attempts to diagnose net effective mixing

in ocean general circulation models is work in progress (Burchard, 2012; Getzlaff et al., 2010, 2012; Ilicak et al., 2012) and, as suggested by (Dietze and Oschlies, 2005), an increasing number of measurements of the saturation state of noble gases in the world ocean may eventually give guidance on the question of realism of simulated diapycnal mixing. For now, however, we do not know the values for diapycnal diffusivity that are to be set explicitly in Earth System Models. Typical choices range roughly between 0.1 to 0.5 $cm^2/s$. Changes from one value to another have been shown to profoundly change simulated dy-

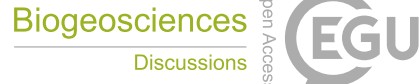



namics of biogeochemical processes, both for historical atmospheric $CO_2$ concentrations and for projections into a warming future (Oschlies, 2001; Duteil and Oschlies, 2011) - when the biogeochemical model is unchanged.

Thus, typically, diapycnal mixing of a specific ocean model must have a profound impact on the respective biogeochemical component and it's parameter settings. The latter holds particularly as the development of Earth System Models is modular and

module after module is coupled together. Generally, the pelagic biogeochemical module is added to an already-coupled ocean-atmosphere kernel. Thus, pelagic biogeochemistry modules are developed within a rather "fixed" physical model environment. This approach is equivalent to assuming that any model-data misfit of biogeochemical cycles is attributable to either a deficient biogeochemical model formulation (i.e., inapt set of partial differential equations), or an inapt choice of biogeochemical model parameters (such as growth rates or limitation thresholds), or both – if the biogeochemical model is tuned (by tweaking equa-

tions and parameters) until "reasonable" agreement with climatological observations (such as nutrients und surface chlorophyll a) is achieved (e.g., Illyina et al., 2013; Keller et al., 2012, and many more to follow).

To summarize: on the one hand, it is well known that the choice of diapycnal diffusivity profoundly affects model solutions and that the value of diapycnal diffusivity, that is to be set in models, is highly uncertain. On the other hand, it is common practice to tune biogeochemical modules to a fixed physics (which is difficult to evaluate). We conclude that this practice

entails the danger of what we coin *reciprocal bias compensation*, where the flaws of one model component (ocean mixing) are compensated by tuning/tweaking another model component (biogeochemical mixing). The final result might be two flawed model components (sometimes without even the modellers knowing).

This study sets out to illustrate the effect of reciprocal bias compensation by replicating the typical workflow of Earth System Model development with *twin experiments*, based on the University of Victoria Earth System Climate Model (UVic Weaver

et al., 2001): we define the Keller et al. (2012) model simulation as the *Genuine Truth*. Further, we define a twin that has a biased ocean component relative to the Genuine Truth in that the vertical diffusion is increased severalfold (configuration MIX+). Finally, we optimize the biogeochemical parameters of the biased twin such that it resembles the Genuine Truth as closely as possible (configuration TUNE). Such an approach gives us full control over the abundance of data (model output) in space and time and the underlying equations (which is not the case for real world observations). In a nutshell, our study

discusses three model configurations, applied to both, historical and anticipated future atmospheric $CO_2$ concentrations. The model setups comprise a Genuine Truth, a biased version of the Genuine Truth and a reciprocally bias compensated version of the latter (cf. Tab.2).

We describe the design of the numerical experiments in detail in the following section. Section 3 compares model results driven with historical and anticipated future atmospheric $CO_2$ concentrations (RCP8.5 scenario). Section 4 discusses the results

and Section 5 closes with a summary and conclusions.

## 2    Methods

This study is based on a suite of numerical experiments performed with the University of Victoria (UVic) Earth System Climate model (version 2.9), a model of intermediate complexity with relatively low computational cost. The model has recently been



applied to explore geo-engineering options in a number of studies (e.g. Keller et al., 2014; Oschlies et al., 2010; Matthews et al., 2008; Reith et al., 2016). In the following Sections 2.1 and 2.2 we present the three configurations of the model used in this study. Two configurations are very similar and one configuration is identical to the configuration that has been comprehensively documented and assessed in the model description paper by Keller et al. (2012).

## 2.1 Earth System Model

Common to all of our three UVic configurations is a horizontal resolution of $1.8°$ in latitude and $3.6°$ in longitude in all submodules (land with active terrestrial vegetation component, ocean, atmosphere, dynamic-thermodynamic sea ice, simple land ice; Weaver et al., 2001). The atmospheric component comprises a single-level atmospheric energy-moisture balance model. Surface winds are prescribed from the NCAR/NCEP monthly climatology. The prescribed winds are used to calculate the momentum transfer to the ocean, the momentum transfer to a dynamic/thermodynamic sea ice model, the surface heat and water fluxes, and the advection of water vapor in the atmosphere.

The ocean submodule is based on a three-dimensional primitive-equation model Pacanowski (2010). The vertical discretisation of the ocean comprises 19 levels. The vertical resolution increases gradually from $50\,m$ at the surface to $500\,m$ at depth. The vertical background mixing parameter, $\kappa_h$ is constant ($0.15\,cm^2s^{-1}$ in the reference version) - apart form the Southern Ocean (south of $40°S$) where the background value is increased by a value of $1.0\,cm^2s^{-1}$. An anisotropic viscosity scheme from Large et al. (2001) is implemented, as in Somes et al. (2010), to improve the equatorial circulation.

A marine pelagic biogeochemical model is coupled to the ocean circulation component. Its prognostic variables are phytoplankton ($P_o$), diazotrophic phytoplankton ($P_D$), zooplankton (Z), detritus (D), nitrate ($NO_3$), phosphate ($PO_4$), dissolved oxygen ($O_2$), dissolved inorganic carbon (DIN) and alkalinity (ALK). The original configuration of Keller et al. (2012) has been tuned tuned to match the annual mean nutrient fields provided by the World Ocean Atlas (Garcia et al., 2010).

The temporal evolution of each prognostic variable is given by

$$\frac{\partial C}{\partial t} = T + S, \tag{1}$$

where T denotes the convergence (or divergence) of physical transports (sum of advection, isopycnal and diapycnal diffusion) and S denotes the source minus sink terms (such as differences between growth and death, air-sea fluxes, sinking). The model is described in detail in Keller et al. (2012) and Schmittner et al. (2008). Here, we present a choice of details relevant to those processes that we change (in one of the configurations, as described in Section 2.2). These processes are the sinking of detritus, the remineralization of detritus and grazing by zooplankton. More specifically we apply changes to the following model parameters (see also Table 1): (1) the sinking speed of detritus $w_{D0}$, (2) the remineralization rate of detritus $\mu_{D0}$ and the maximum grazing rate $\mu_Z$. In the following we present the equations in which these model parameters are applied.

Phytoplankton growth is controlled by the availability of light and nutrients (here, nitrate, phosphate and iron, where the latter is parameterized by an iron mask rather than explicitly resolved). Phytoplankton blooms are terminated by zooplankton grazing once essential nutrients are depleted. Zooplankton grazes on phytoplankton, diazotrophs, detritus and other zooplankton (self-predation). Zooplankton growth is limited by a maximum zooplankton growth rate. This rate, $\mu_Z^{max}$, is dependent on



temperature (T) up to 20°C and oxygen concentrations (O$_2$ in units $mmol\,m^{-3}$). In one of our configurations we change the value of the model parameter $\mu_Z$ in the equation

$$\mu_Z^{max} = \mu_Z\,max(0,(0.5(\tan(O_2-8)+1)1.066^{min(20,T)}))$$

Both, phytoplankton and zooplankton produce detritus which sinks to depth. This sinking speed, in combination with the
remineralization rate determines the depth at which detritus is converted back into dissolved species (such as nitrate, phosphate, DIC) and thus ends its route down to the sea-floor. The sinking speed of detritus, $w_D$, increases linearly with depth:

$$w_D = w_{D0} + m_W z,$$

where $w_{D0}$ is the detritus sinking speed at the surface, $m_W$ denotes depth dependent detritus sinking speed and $z$ is the effective vertical coordinate (positive downward). In one of our configurations we change the value of the model parameter $w_{D0}$.
Remineralization of detritus returns the nitrogen (N) and phosphorus (P) content of detritus back to NO$_3$ and PO$_4$, consumes oxygen and releases inorganic carbon. The rate of remineralization $\mu_D$ is both temperature-dependent (T) and a function of ambient oxygen concentrations (it decreases by a factor of 5 in suboxic waters):

$$\mu_D = \mu_{D0}\,exp(\frac{T}{T_b}(0.65+0.35\tanh(O_2-6))).$$

$T_b$ is the e-folding temperature of biological rates (notation after Schmittner et al., 2008), $T$ and $O_2$ are ambient temperature
(°C) and oxygen concentration ($mmol\,O_2\,m^{-3}$), respectively. In one of our configurations we change the remineralization rate $\mu_{D0}$ which sets the rate at 0°C under oxygen replete conditions.

## 2.2 Numerical Experiments

We present results from numerical experiment of three different configurations of the numerical Earth System Model UVic (cf., Tab.2). For all three different configurations we apply both, constant atmospheric CO$_2$ concentration as well as increasing
atmospheric CO$_2$ concentrations over time.

### 2.2.1 Model Configurations

Table 2 lists the model configurations. The *Genuine Truth* configuration is identical to the *Reference Simulation* in Getzlaff and Dietze (2013) and has been developed and introduced by Keller et al. (2012). Note that this *ground truth* model version by Keller et al. (2012) was modified (or tuned) such that the misfit to climatological observations of biogeochemical relevance,
such as dissolved phosphate concentrations and phytoplankton (Garcia et al., 2010), is reduced relative to the original biogeochemical module from Schmittner et al. (2008). We define this model configuration as ground truth (*Genuine Truth*) to which we compare the model configurations with increased mixing, MIX+ and TUNE.

The first model modification, referred to as MIX+, is identical to the model version underlying the Genuine Truth, except for an increase in the vertical background diffusion from $\kappa_h$= 0.15 up to 0.4 $cm^2 s^{-1}$ (cf., Table 2 and 3). This choice is motivated
by Kvale et al. (2017) who increased $\kappa_h$ from 0.15 to 0.43 $cm^2 s^{-1}$ (in the same model) in order to compensate for a collapsed





meridional overturning circulation when switching from one numerical advection scheme to another. Also the regarded value is well within the range explored by Duteil and Oschlies (2011).

TUNE is another twin to the Genuine Truth and identical to MIX+, except for changes to three biogeochemical model parameters: (1) sinking speed of detritus at the surface, $w_{D0}$, (2) remineralization rate of detritus, $\mu_D$, and (3) maximum

zooplankton growth rate, $\mu_Z$ (cf., Table 3). The leading thought behind these changes relative to MIX+ is to mimic the behavior of the Genuine Truth configuration even though the vertical background diffusion is substantially different to the Genuine Truth. Or, in other words, changes to biogeochemical model parameters in TUNE are chosen such that the bias between the Genuine Truth and MIX+ (that is effected by the choice of a higher background diffusivity) is compensated. The procedure to achieve such a bias compensation is as follows: (1) We chose the three parameters somewhat arbitrarily guessing that they are capable

of reciprocally compensating for the effect of an increased vertical diffusivity. (2) We performed 50 spinups (cf., Section 2.2.2) with increased diffusivity and differing sets of values for the aforementioned biogeochemical model parameters. (3) From this set of 50 we chose the configuration TUNE which was "most similar" to the Genuine Truth. Following the rather generic workflow of biogeochemical model development we defined "similar to the Genuine Truth" as yielding a low root-mean-square deviation with respect to surface chlorophyll a and oceanic phosphate concentrations (both values were averaged after unit

conversion via a fixed Redfield-ratio). This comparison between between TUNE and the Genuine Truth was performed under preindustrial atmospheric $CO_2$ concentrations. (Note that there is an ongoing discussion on misfit metrics which is beyond the scope of this manuscript, e.g., Evans, 2003; Löptien and Dietze, 2015). Please note that parameter choice for TUNE that yield an even better bias compensation than the one we present in Table 3 may well exist and be found by using automated parameter optimization approaches such as suggested by Sauerland et al. (2009). Also, the bias might potentially be lowered

when considering more biogeochemical model parameters. However, given the already-remarkable similarity between TUNE and the Genuine Truth (as will be put forward in Section 3), we decided against the associated computational cost for the rather illustrative purpose of this study.

### 2.2.2   Spinup Procedure, Historical Model Solution, and Projections into the Future

All numerical experiments presented in this study start from observed tracer distributions (Garcia et al., 2010). Each of the three

model configurations (Genuine Truth, MIX+ and TUNE, Section 2.2.1) is then integrated under pre-industrial atmospheric $CO_2$ concentrations for 3000 years in order to reach quasi-equilibrated spun-up model states (e.g. Gupta et al., 2013) for all three configurations. The results (more specifically, the average of the last 10 years of the respective 3000 year spinups) of these three numerical experiments are dubbed *Historical Model Solutions* because they are representative for a pre-industrial world.

Subsequently, starting from the respective spun-up states of the three model configurations, so-called 1000 year-long drift

runs are performed, where the atmospheric $CO_2$ content is allowed to vary in response to pre-industrial emissions. After this drift phase also virtual air-sea fluxes of biogeochemical species are turned on (i.e., changes in DIC due to evaporation, precipitation and runoff - Weaver et al. (2007)). This procedure has proven to be efficient in switching from a prescribed atmospheric $CO_2$ setup to an atmospheric emission-driven setup while keeping the spinup-times within reasonable bounds



(also used in e.g. Löptien and Dietze, 2017). Finally, we annex projections into the future by considering the years 1850-2100. From 2005-2100 we apply the emission scenario RCP 8.5 (Riahi et al., 2011).

In summary, we present 6 numerical experiments: a historical model solution and a RCP 8.5 scenario for each of the three model configurations - Genuine Truth, MIX+ and TUNE.

## 3 Results

In the following Section 3.1 we present results from the three historical model simulations, based on the model configurations Genuine Truth, MIX+ and TUNE, respectively. This subsection illustrates the potential of *reziprocal bias compensation*. In Section 3.2 we present the results from the RCP 8.5 scenario simulations. The aim of the latter is to explore the robustness of the reziprocal bias compensation under a typical global warming scenario.

### 3.1 Historical Model Solutions

The massive, severalfold, increase in background diffusivity introduces surprisingly litte differences in simulated ocean physics: the differences in sea surface temperature are barely visible in Figure 1. A similar picture evolves for sea surface salinity where, however, upon closer inspection of Figure 2, a minor increase in sea surface salinity in the Southern Ocean and in the Arctic in response to increased vertical mixing rates is visible. This is consistent with an increase in the meridional overturning circulation from $19\,Sv$ in the Genuine Truth to $22\,Sv$ in Mix+, which compensates for some of the net air-sea freshwater fluxes in the high latitudes (and thus increases sea surface salinities in these latitudes). Expressed in terms of a global mean difference, the Genuine Truth and MIX+ historical simulations differ by $0.03\,K$ and $0.13\,PSU$ only. Figure 3 supports the impression of similarity by showing that differences in the simulated zonally averaged net air-sea heat fluxes are within the range of measurement uncertainty in the field (e.g. Gulev et al., 2007). High values are restricted to very limited regions, impacted by sea ice or deep water formation.

In contrast to the barely detectable changes in physical ocean dynamics, described above, conventional proxies of biogeochemical cycling turn out to be very sensitive to the change in vertical background diffusivity. The surface phosphate concentrations (compare left and right panel of Figure 4) and surface phytoplankton concentrations (compare left and right panel of Figure 5) showcase this amplified sensitivity of biogeochemical cycling to changes in governing physics in a drastic way: while the increased vertical diffusion of cold abyssal waters from depth to the surface effects only minor changes to sea surface temperature and air-sea heat fluxes (as discussed above), it brings substantially more phosphate to the nutrient-depleted sun-lit surface layer where it drives a substantial increase in the autotrophic production and standing stock of phytoplankton. As a consequence, the cycling of phosphate in the upper thermocline accelerates. The export of particulate organic matter increases and the subsequent remineralization sharpens the vertical nutrient gradient at the base of the euphotic zone which, in turn, increases diffusive nutrient fluxes to the sun-lit surface layer. Among the overall net-effects is the increased phosphate pool in the upper thermocline, shown in Figure 6 (except for in the Southern Ocean where the combination of iron limitation, seasonal light limitation and unique ventilation patterns overcomes the aforementioned effect). As concerns dissolved oxygen,



the effects of increased vertical diffusivity are antagonistic. On the one hand, the increased mixing ventilates the abyssal ocean by mixing oxygenated surface waters downward. On the other hand, the increased mixing accelerates biogeochemical cycling of organic matter (as described above) and thus, as a consequence of the associated accelerated remineralization of organic matter, increases the oxygen demand. Figure 7, right panel, reveals that the ventilating effect prevails in MIX+, i.e., the oceanic

oxygen inventory rises in response to the higher diffusivity.

The above results are inline with the intended model design (which mimics the typical workflow of Earth System Model development): The Genuine Truth simulation represents a global set of (synthetic) observations. MIX+ is a physically biased model version of the Genuine Truth with, as illustrated above, drastic consequences for the simulated biogeochemical tracer distributions. The setup TUNE is an attempt to tune/tweak the biogeochemistry in the biased model MIX+ such that it resembles

the Genuine Truth under historical conditions. Thus in TUNE we compensate the biased imposed in the physics by tuning the biogeochemistry. The results under historical forcing are as follows: In terms of physical ocean circulation TUNE and MIX+ are almost identical (not shown). The main reason for this insensitivity is that the atmospheric $CO_2$ concentrations are prescribed to the same preindustrial levels in both of the historical solutions. Hence the feedback from changed biogeochemistry via oceanic carbon uptake to atmospheric $CO_2$ and associated changes in air-sea heat fluxes is excluded. In terms of biogeochemistry TUNE

and MIX+ do differ considerably. Figure 4 and Figure 5 show that the surface phosphate and phytoplankton concentrations simulated with TUNE are much more alike the Genuine Truth than MIX+. The similarity between the Genuine Truth and TUNE (in contrast to the difference between the Genuine Truth and MIX+) is not restricted to surface properties but extends to depth. For example, Figure 6 shows that the zonally averaged phosphate concentrations of TUNE are much more similar to the Genuine Truth than is the case with MIX+. Further, Figure 7 shows a similar behaviour of the oceanic oxygen inventory.

The latter holds as well for the global extent of the simulated suboxic volume (not shown). The latter is remarkable since oxygen was not included into our metrics applied for the tuning process. Table 4 provides a more quantitative measure of the bias compensation: the root-mean-square deviation between global distributions of phosphate (phytoplankton) concentrations simulated with the Genuine Truth and MIX+ is $0.16\,mmol\,P\,m^{-3}$ ($0.033\,mmol\,N\,m^{-3}$). By tweaking biogeochemical model parameters in simulation TUNE this bias is reduced by $\approx 40\%$ (down to $0.1\,mmol\,P\,m^{-3}$ and $0.021\,mmol\,N\,m^{-3}$, Table 4).

## 3.2  Model Projections into a Warming Future

All of our numerical configurations agree in that they feature a considerable sea surface temperature increase by the year 2100 when driven by the RCP8.5 greenhouse gas emission scenario (Riahi et al., 2011). The associated increase in radiative forcing warms the surface ocean and increases the stability of the water column (because relatively warmer and more buoyant water sits on top of cold abyssal water).

Expressed in terms of a global mean sea surface temperatures, the projected increase differs by up to 12% depending on the underlying model configuration: the projected global mean temperature rise is $2.5\,K$, $2.2\,K$, and $2.3\,K$ for the configurations Genuine Truth, TUNE and MIX+, respectively. These differences among the experiments are consistent with the fact that (by construction) the simulations based on MIX+ and TUNE distribute the heat over greater depth. Thus, their increased background diffusivity cools the surface (and warms the deep ocean) relative to the Genuine Truth. Consequentially, TUNE





and MIX+ feature a stronger warming in the deep ocean than the Genuine Truth (doubled below 1500 m depth). This effect is somewhat offset in MIX+ which shows more temperature increase at the surface than TUNE, even though MIX+ and TUNE share the same physical model parameters. This is consistent with MIX+ featuring a phytoplankton standing stock which exceeds 150% relative to both, TUNE and the Genuine Truth. This increased phytoplankton concentration absorbs more light

at the surface and intensifies the surface warming (relative to TUNE and the Genuine Truth where more light is absorbed at depth below the surface).

Regionally, the differences in the projected sea surface temperatures (SSTs) are much larger than in the global mean: Figure 8 shows a comparison of sea surface temperature warming between the Genuine Truth and TUNE configuration in response to the RCP8.5 emission scenario. The differences between the Genuine Truth and TUNE configuration exceed at max. 1.8 $K$. The

overall pattern is 0.2 to 0.5 $K$ more warming of the Genuine Truth in the Northern Hemisphere and 0.1 to 0.5 $K$ less warming in the Southern Ocean. Hence, Southern Ocean SST warming in TUNE in response to increased greenhouse gas emissions (RCP8.5 scenario) is stronger than in the Genuine Truth, even though the Genuine Truth, overall, warms quicker than TUNE (as put forward in the previous paragraph). We speculate that the increased background diffusivity in TUNE reduces the cooling effect of deep convection in the Southern Ocean by 2100 (relative to the Genuine Truth) because the abyssal waters (into which

the deep convection taps into) in TUNE have received more heat (relative to the Genuine Truth) prior to year 1850, already. Also the max. overtuning shows a stronger projected decline with increased vertical diffusivity.

In terms of biogeochemistry, the similarity of model projections depends on the considered metric. For some biogeochemical tracers the reciprocal bias compensation (where increase in diapycnal mixing is compensated by changes to biogeochemical model parameters) is robust under global warming while the historical similarities break for others. In the following Subsec-

tions 3.2.1 to 3.2.4 we illustrate the respective range. We start with a metric that is most robust and end with a metric where the bias compensation breaks under the emission scenario.

### 3.2.1 Surface Phytoplankton Concentrations

Projections of phytoplankton are of interest also because phytoplankton forms the base of the food chain and thereby exerts control on fisheries. The Genuine Truth simulation projects globally decreasing surface phytoplankton concentrations (cf.,

Figure 9 left panel, corresponding to a global mean decrease of 7% by 2100). This is consistent with the increased stability of the water column (effected by global warming) reducing the turbulent vertical mixing of nutrient replete waters from depth to the nutrient-depleted sun-lit surface ocean. In limited regions the projected changes can be counter to the overall trend. These differences are most likely attributed to circulation changes (cf., Fig.8). Examples of such regions are the Arctic, the equatorial Pacific and the Southern Ocean.

The projection based on MIX+ differs substantially from the Genuine Truth in that it projects an overall increasing surface phytoplankton concentration (cf., Figure 10). Most of this difference to the Genuine Truth is agglomerated in the Pacific Ocean, a region infamous for it's nonlinear behavior (in our model, cf., Section 3.2. in Löptien and Dietze, 2017). The projected phytoplankton change based on TUNE is very similar to the Genuine Truth (cf., Figure 9, middle panel), corresponding to a



globally-averaged decrease of 8% by 2100 - even though it shares the same biased physics with MIX+. This illustrates that for projected surface phytoplankton patterns the reciprocal bias compensation is (in our model) robust under the RCP8.5 scenario.

### 3.2.2 Surface Phosphate Concentrations

Projections of phosphate are of interest also because it is an essential nutrient that limits the growth of phytoplankton and the

associated biotic export of organic matter to depth. In most state-of-the-art global coupled ocean-circulation biogeochemical models phosphate is the "base currency", i.e., its cycling is directly (often linearly) related to the cycling of plankton and gases such as oxygen and $CO_2$. The Genuine Truth simulation projects globally decreasing surface phytoplankton concentrations (cf., Figure 10, left panel), corresponding to a global mean decrease of 17% by 2100. As described above, this result is consistent with the increased stability of the water column, that is effected by net air-sea heat fluxes (and associated buoyancy

because warmer water is lighter than colder) caused by global warming. In the Southern Ocean, processes at work are more complex. Here, the projected sea surface temperatures in the Genuine Truth simulation show alternating pattern of increasing and decreasing sea surface temperatures until the year 2100 (cf., Fig. 8). Downstream of regions where sea surface temperatures are reduced, more nutrients are mixed up to the surface in convective events and, simultaneously, surface mixed layers are increased. The latter leads to opposite effects, compared to the above considerations on global scale.

The MIX+ projection is similar to the Genuine Truth in that it projects that decreasing surface phosphate concentrations prevail globally (cf., Figure 10, right panel), corresponding to a globally-averaged decrease of 17% by the year 2100. In the Southern Ocean, however, MIX+ differs substantially in that the alternating patches of increasing and decreasing surface phosphate concentrations apparent in the Genuine Truth are smoothed out (Figure 10, compare left and right panel). We speculate that, the absence of patches with strongly-increasing surface phosphate indicates less deep convection events. The

latter is inline with the projected SSTs (Fig. 8).

The projection based on TUNE is generally very similar to the MIX+ projection: decreasing surface phosphate concentrations prevail globally, consistent with the Genuine Truth (cf., Figure 10, middle panel), corresponding likewise to a globally-averaged decrease of 17% by 2100. In the Southern Ocean, however, TUNE is much more similar to MIX+ than the Genuine Truth. This illustrates that for projected surface phosphate patterns the reciprocal bias compensation is not very robust (espe-

cially locally, where circulation effects kick in) and some of the similarities apparent under historical conditions break under the RCP 8.5 emission scenario.

### 3.2.3 Total Oceanic Carbon Content

Projections of the oceanic carbon content are of interest also because the oceans currently take up a significant fraction of the anthropogenic carbon emissions (of the order of 25%, e.g. Takahashi et al., 2002). To this end changes in the capability of the

ocean to sequester carbon away from the atmosphere in a warming future will directly affect the rate of global warming itself. This strong feedback is among the main drivers behind the development/inclusion of biochemical carbon modules in Earth System Models that are used to assess the effects of climate change (and to develop mitigation strategies).



Figure 11 shows that the reziprocal bias compensation is not robust when regarding the projected oceanic carbon content: Historically, the ocean simulated carbon content of the ocean between the simulations varies by less then 0.5% between the simulations. Still, both MIX+ and TUNE propose to take up $200\,GTC$ more than the Genuine Truth by the year 2100. In line with earlier studies, we presume that the latter must be attributed to differences in the solubility pump. Our results indicate that

for oceanic carbon content the reciprocal bias compensation is not robust once the boundary conditions strongly change.

### 3.2.4   Suboxic Volume

Projections of suboxic volume are of interest also because suboxia trigger denitrification and thus reduce the global availability of fixed nitrogen which is an essential nutrient for all phytoplankton other than diazotrophs. Figure 11 shows that the suboxic volume, according to the Genuine Truth projection, decreases with global warming. A similar surprising behavior has been

reported from other Earth System Models (e.g. Gnanadesikan et al., 2011) – surprising since it is counter to intuition because (1) warming reduces the solubility of oxygen and (2) the increased stratification of the water column which comes along with stratification reduces ventilation, i.e., both of these directly-to-warming associated processes tend to reduce oceanic oxygen levels and thus promote suboxia.

MIX+ shows that an increased background diffusion can reverse the projected trend in suboxic volume. This is consistent

with results from Gnanadesikan et al. (2011) and Getzlaff and Dietze (2013) highlighting the sensitivity of suboxic waters to the resolved and parameterized ocean physics.

The projection based on TUNE behaves like MIX+ in that it also shows a trend opposing the Genuine Truth. Remarkably its trend is even more off relative to the Genuine Truth than the trend of MIX+. This illustrates that for suboxic volume the reciprocal bias compensation is not at all robust when the models are projected into the future.

## 4   Discussion

We set out to explore *reciprocal bias compensation* in Earth System Models where biases in the ocean circulation module are deliberately compensated by biases in the biogeochemical module. In the following, we will discuss the choice of our Earth System Model (Section 4.1), the changes applied to the physical module (Section 4.2), and the changes applied to the biogeochemical module (Section 4.3). In Section 4.4 we will discuss the similarity between TUNE and Genuine Truth under

historical forcing (and argue, based on observations to-date, that we can not decide which is better suited to make reliable projections into a warming future). Section 4.5 closes this Discussion with highlighting the differences between projections based on the model configurations TUNE and the Genuine Truth.

### 4.1   Choice of model framework

Our results are based on integrations of the UVic Earth System Model (Keller et al., 2012). The model is relatively simple (i.e.,

it is an Earth System Model of Intermediate Complexity (EMIC)) and rather coarsely resolved ($\approx 200km$), compared to the cutting-edge generation of IPPC-type Atmosphere-Ocean General Circulation Models (AOGCMs). We speculate, however, that





since, EMICs and AOGCMs share very similar (or sometimes even identical) ocean circulation and pelagic biogeochemistry kernels, that our EMIC-based results are applicable also to IPCC-type AOGCMs. Accordingly, earlier findings from model intercomparison studies (e.g., Najjar et al. (2007)) are inline with our results. Still we have to stress that our mixing parameter settings can presumably be regarded as extreme cases, chosen to cover the whole range.

We compare three configurations of UVic dubbed the *Genuine Truth, MIX+* and *TUNE*. All setups are all very similar (the Genuine Truth even identical, cf., Section 2.2.1) to the Keller et al. (2012) configuration. This choice is motivated by the fact that this configuration is extensively used to assess the impact of geo-engineering options. Among recent studies are Partanen (2016) who explored the impacts of sea spray geo-engineering, Reith et al. (2016) who explored the effects of carbon sequestration by direct injection into the ocean and Mengis et al. (2016) who assessed the effects of ocean albedo modification
in the Arctic.

## 4.2   Choice of modification to background diffusivity

Two of our configurations (MIX+ and TUNE) feature a slightly changed physics relatively to the original Keller et al. (2012) configuration (which we dubbed Genuine Truth). The choice of changing the vertical background diffusivity is motivated by the fact that vertical diapycnal mixing is not well quantified, neither in models nor in the real ocean. What is known, though, is
that diapycnal mixing is highly heterogenic both in time and space. Enhanced diffusivities up to 10 cm$^2$/s near the bottom have been observed over rough topography (Ledwell et al., 2000), while large scale estimates derived from purposeful tracer release experiments in the subtropical North Atlantic yield values of $0.17 \pm 0.02$ cm$^2$/s when considering a 2 year average (Ledwell et al., 1998). Even if the temporal and spacial variability would be mapped out (which is not yet the case), the challenge is to transfer these numbers into a model parameterization that ensures realistic diffusive transports (of heat, salt, nutrients etc.) -
which are defined as the product of respective spacial (vertical) property gradient and diffusivity. Hence, using a diffusivity that is averaged over time and space (as is inevitable in the current generation of models that apply a finite-differences discretization) but, at the same time, is fraught with high uncertainties. An additional source of uncertainty is implicit diffusion. As opposed to explicitly prescribed diffusion implicit diffusion is a spurious and hard-to-quantify artifact (cf., Getzlaff et al. (2012) ) of the underlying numerical advection algorithm.

To summarize: the uncertainty is substantial. We use the original diffusivity proposed for UVic and compare it to a value that was applied in the same model that uses a slightly different advection scheme but is identical otherwise. The latter change is motivated by the fact that a version with a higher, of similar order, background diffusivity is used as the base for an offline approach which saves considerable computational cost. The visible effects on physical tracers when using ad-hoc measures, based on temperature and salinity, are rather weak. In terms of global sea surface temperature differences only 0.03K. In terms
of meridional averaged heat fluxes the differences are below 5W/m$^2$ form 50°S to 50°N. At high latitudes differences reach 25W/m$^2$.

    The max. meridional overturning circulation increases (as expected) with increased vertical diffusion and is 22 $Sv$ in the reference simulation of MIX+ versus 19 $Sv$ in the Genuine Truth. To rate this numbers, we put them into perspective with IPCC models: Marsland (2013) show values in the range 22 -24 $Sv$ for the Australian Community Climate and Earth System



Simulator coupled CMIP5 model under pre-industrial conditions. For the late twentieth century, Reintges et al. (2017) report an ensemble mean of a max. overturning of 19 $Sv$ in CMIP5 models. The spread among models in the late twentieth century is with 21 $Sv$ huge, and even exceeds the spread of the simulated max. meridional overturning in CMIP3 models. Thus, the difference between the lowest and highest projected max overturning in CMIP5 models is even higher as the present day

observational estimate (17.5 $Sv$ based on the RAPID array at 26°N, Smeed et al. (2014)).

### 4.3 Choice of changes to biogeochemical model parameters

One of our model configurations TUNE has both, changed physics, like MIX+ (described above), and changed biogeochemisty. In TUNE biogeochemical parameters $\mu_Z$, $\mu_D$ and $w_{D0}$ have been changed by 12.5, ≈200 and 115% respectively. These changes are well within uncertainties, i.e., within the range of what is applied in other studies. E.g. Kriest et al. (2017) assumes

that the zooplankton mortality is uncertain within 4000% and the sinking speed at the surface in the Genuine Truth setup exceeds the value used by Illyina et al. (2013) by 180%. Also, all considered parameter values are well within the range used by Schartau (2003) in an automated parameter optimization study.

### 4.4 Similarity between the reciprocally bias compensated couple

For the historical runs we showed in Section 3.1 that the simulations TUNE and the Genuine Truth are very similar to one

another, i.e., that the reciprocal bias compensation was pretty effective. Given the close resemblance of temperature, salinity, air-sea heat fluxes, phosphate, phytoplankton, and oxygen in Fig. 1 to 7, we argue that both simulations, TUNE and the Genuine Truth, feature comparable misfits to observations for common model evaluation metrics. Since the Genuine Truth configuration is considered state-of-the-art (here meaning to be good enough to carry out geo-engineering studies as outlined above) we conclude, in-turn, that the configuration TUNE can also be regarded as state-of-the-art.

### 4.5 Ensuing uncertainties in response to RCP8.5

As outlined above, it is hard to argue based on a-priori knowledge as concerns the differences among their underlying model parameters, which model configuration, the Genuine Truth or TUNE, is more realistic. Also choosing the better model based on their performance in reproducing historical observations is difficult: the differences are rather small and the best choice will presumably change with the applied metric (or cost) to measure the misfit to the observations. Still, it is somewhat disconcert-

ing, that the two configurations do differ in what they project to come in response to the RCP8.5 emission scenario:

   In terms of the uncertainties in projected sea surface temperatures imposed by ocean mixing are locally substantial: for the northern hemisphere, we found differences of on average 0.5 $K$ between the Genuine Truth and the simulation TUNE. This difference is substantial, particularly given that consensus has been reached trying to keep global warming below 2 $K$. This finding, that vertical diffusion matters, is inline with earlier studies, that stress the importance of the vertical diffusion

coefficient on several physical properties of the ocean: e.g., Bryan (1987) pointed out that this parameter impacts strongly the



sensitivity towards wind forcing and Bryan (1987) stresses the importance for the simulated large-scale meridional overturning circulation.

In line with results from, e.g., by Oschlies (2001) and Duteil and Oschlies (2011), we find that the uncertainty in background diffusivity maps as well onto uncertainties in projected biogeochemical tracers. Striking examples, put forward in Section 3.2,

are phytoplankton concentrations in the equatorial Pacific and suboxic volume (where projections based on TUNE and the Genuine Truth do not agree even on the sign on projected changes). The results for suboxic volume is consistent with findings by Cabré et al. (2015) who illustrate (in their Figure 9) that the current generation of CMIP5 models does not agree on the sign of change either.

In terms of oceanic carbon content, the differences between TUNE and the Genuine Truth in projected future changes

accumulate to 200 Gt of carbon in the year 2100. This difference is substantial: Expressed in terms of today's anthropogenic carbon emissions, the difference corresponds to 20 years worth of anthropogenic emission which covers more than 40% of the respective differences among CMIP5 models (Friedlingstein et al., 2014).

## 4.6 Model evaluation metrics

Our study stresses the urgent need to evaluate the mixing in ocean models carefully before projecting into the future. With

this our results are inline with earlier findings within the Ocean Carbon-cycle Model Intercomparison Projects (OCIMIPs), that already early on highlighted the importance of a realistic representation of the physical ocean processes for modelling the pelagic biogeochemistry (e.g., Doney et al. (2004)). A definition of a suited evaluation metric, apart from assessing relatively simple common measures, such as temperature, salinity and MOC-strength, is still not straightforward and today there is no consensus how to rate ocean models. It was suggested to consider also ventilation times (e.g., Matsumoto et al. (2004)). Ac-

cordingly, additional inert chemical tracers (e.g., chlorofluorocarbons (CFC11, CFC12), $SF_6$), allowing for an extended model evaluation, are now required as standard output within CMIP6 (Orr et al. (2017)). Still, many open questions and problems remain with such approaches. For example the parameterized air-sea gas exchange induces large uncertainties (e.g., England et al (1994)). Also, dating ranges of CFCs are not suitable to resolve the dynamics of the deep ocean which recently lead to the suggestion to introduce additionally $^{39}$Ar (Ebser et al, 2018). The latter, promising approach is currently under investigation. In

summary, however, it still still remains an pressing open question which misfit metrics ensure reliable projections. One major aim of the presented study is thus to remind on this perpetual, often disregarded problem and to trigger related work.

## 5 Summary and Conclusions

We present results from two configurations of an Earth System Model that feature a very similar behavior when driven with historical forcing but diverge drastically when used to project into our warming future, using the anthropogenic green house

gas emission scenario RCP8.5. The difference between the two configurations (dubbed the Genuine Truth and TUNE) are a modified vertical background diffusivity and changes applied to biogeochemical model parameters. The respective modification of the vertical diffusivity is within the range of what has been recently applied (cf., settings in Reith et al., 2016; Kvale

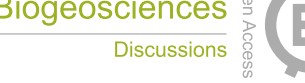



et al., 2017). Likewise, the changes in biogeochemical model parameters are within the envelope of state-of-the-art model configurations (Schartau, 2003).

In terms of typical (referring to currently-applied assessment metrics such as in Kvale et al., 2017) physical (such as historical sea surface temperatures and meridional overturning estimates) and biogeochemical metrics (such as historical observations of
phytoplankton and phosphate concentrations) none of our two model configurations can be favored or discarded. The reason is that, for one, our increase of vertical diffusivity has rather little effects on the physics in our model framework. Second, while the effect of our increased diffusivity substantially offsets generic biogeochemical assessment metrics (cf. configuration MIX+) we were able to compensate for most of the respective bias by changing biogeochemical model parameters.

When driven with a RCP 8.5 scenario, however, the similarity between our model configurations brakes for projections of
societal relevance, such as the oceanic uptake of carbon or the dynamics of oxygen minimum zones. For carbon the projections accumulate a difference of 20 years worth of today's anthropogenic emissions by the end of 2100. For the suboxic volume, not even the sign of forecasted changes coincides.

We conclude that an improved understanding of vertical diapycnal mixing in Earth System Models alleviates the risk of reciprocal bias compensation by (wrongly) tweaking biogeochemical modules to a deficient physics.
Thus an improved understanding of vertical diapycnal mixing can be expected to reduce ensuing uncertainties in climate projections considerably. Reverse reasoning suggests that an improved understanding of biogeochemistry can help to assess the realism of diapycnal mixing in Earth System Models because: (1) we found that some biogeochemical metrics are more sensitive to changes in mixing parameterization than typical physical metrics (such as overturning and temperature distributions), (2) if the biogeochemical model formulations could be sufficiently constricted then misfits between (more sensitive) biogeo-
chemical metrics can be related back to deficiencies in the physical component of coupled ocean-circulation biogeochemical models. In summary, our studies illustrates the need for suitable model assessment metrics to reliably project into the future.

## 6   Conclusions

We present results from two configurations of an Earth System Model that feature a very similar behavior when driven with historical forcing but diverge drastically when used to project into our warming future, using the anthropogenic green house
gas emission scenario RCP8.5. The difference between the two configurations (dubbed the Genuine Truth and TUNE) are a modified vertical background diffusivity and changes applied to biogeochemical model parameters. The respective modification of the vertical diffusivity is within the range of what has been recently applied (cf., settings in Reith et al., 2016; Kvale et al., 2017). Likewise, the changes in biogeochemical model parameters are within the envelope of state-of-the-art model configurations (Schartau, 2003).
In terms of typical (referring to currently-applied assessment metrics such as in Kvale et al., 2017) physical (such as historical sea surface temperatures and meridional overturning estimates) and biogeochemical metrics (such as historical observations of phytoplankton and phosphate concentrations) none of our two model configurations can be favored or discarded. The reason is that, for one, our increase of vertical diffusivity has rather little effects on ad hoc measures of the physics in our model frame-





work. Second, while the effect of our increased diffusivity substantially offsets generic biogeochemical assessment metrics (cf. configuration MIX+), we were able to compensate for most of the respective bias by changing biogeochemical model parameters. When driven with a RCP 8.5 scenario, however, the similarity between our model configurations brakes for projections of societal relevance, such as the oceanic uptake of carbon or the dynamics of oxygen minimum zones. For carbon the projections

accumulate a difference of 20 years worth of today's anthropogenic emissions by the end of 2100. For the suboxic volume, not even the sign of forecasted changes coincides.

We conclude that an improved understanding of vertical diapycnal mixing in Earth System Models alleviates the risk of reciprocal bias compensation by (wrongly) tweaking biogeochemical modules to a deficient physics, particularly, when using ad-hoc measures to assess the quality of the underlying ocean model. These results are inline with earlier findings within the

Ocean Carbon-cycle Model Intercomparison Projects (OCIMIPs), that highlighted the importance of a realistic representation of the physical ocean processes for modelling the pelagic biogeochemistry (e.g., Doney et al. (2004)). Thus an improved understanding of vertical diapycnal mixing can be expected to reduce ensuing uncertainties in climate projections considerably. Reverse reasoning suggests that an improved understanding of biogeochemistry can help to assess the realism of diapycnal mixing in Earth System Models because: (1) we found that some biogeochemical metrics are more sensitive to changes in

mixing parameterization than typical physical metrics (such as overturning and temperature distributions), (2) if the biogeochemical model formulations could be sufficiently constricted then misfits between (more sensitive) biogeochemical metrics can be related back to deficiencies in the physical component of coupled ocean-circulation biogeochemical models. With these findings our study reminds on the need to develop and apply routinely misfit metrics which take diapycnal mixing into account to obtain more reliable future projections.

*Code availability.* The model code is archived at
https://nextcloud.ifg.uni-kiel.de/index.php/s/PQgtKJpMjr7DTPp.
Since the use of the source code requires registration with the UVic model community (http://climate.uvic.ca/model/), the respective file is password protected and available upon request. All required input files are downloadable under the above link.

*Data availability.* The model output is archived at

https://nextcloud.ifg.uni-kiel.de/index.php/s/PQgtKJpMjr7DTPp

*Author contributions.* Both authors were involved in the design of the work, in data analysis, in data interpretation and in drafting the article.

*Competing interests.* No competing interests are present.





*Acknowledgements.* This work is a contribution of the project "Reduced Complexity Models" (supported by the Helmholtz Association of German Research Centres (HGF)) and the SPP 1158, "Glacial/Interglacial Hydrographic Structures and Nutrient Utilization in the Pacific Southern Ocean - Data and Modeling Approach" (funded by the German Research Foundation (DFG)).



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





**Table 1.** Model parameters explored in this study.

| Parameter | Description | Unit |
|---|---|---|
| $\kappa_h$ | Vertical diffusion | $cm^2 s^{-1}$ |
| $w_{D0}$ | Detritus sinking speed at the surface, | $m\ day^{-1}$ |
| $\mu_D$ | Remineralization rate at $0°$ | $day^{-1}$ |
| $\mu_Z$ | Maximum zooplankton growth rate at $0°$ | $day^{-1}$ |

**Table 2.** Configurations of the numerical earth system model UVic (Weaver et al., 2007) used in this study.

| Configuration | Description | Experiments |
|---|---|---|
| **Genuine Truth** | introduced by Keller et al. (2012), identical to the *reference simulation* Getzlaff and Dietze (2013) | **historical** (+ transition phase) + **RCP8.5 scenario** |
| **MIX+** | identical to the *Genuine Truth* configuration, except for an increase of background diffusivity $\kappa_h$ | **historical** (+ transition phase) + **RCP8.5 scenario** |
| **TUNE** | identical to MIX+ except for changes to the biogeochemical parameters $w_{D0}, \mu_Z, my_{D0}$ | **historical** (+ transition phase) + **RCP8.5 scenario** |





**Table 3.** Parameter values for the three model configurations (cf., Table 2) used in this study (cf., Table 1 for description and units). The Genuine Truth configuration is identical to the one introduced by Keller et al. (2012). MIX+ and TUNE are identical to the Genuine Truth except for the differences in parameter values listed here.

| Configuration | $\kappa_h$ | $w_{D0}$ | $\mu_Z$ | $\mu_D$ |
|---|---|---|---|---|
| Genuine Truth | 0.15 | 12.5 | 0.4 | 0.042 |
| MIX+ | 0.4 | 12.5 | 0.4 | 0.042 |
| TUNE | 0.4 | 40 | 0.45 | 0.09 |

**Table 4.** Misfits of historical model solutions relative to the Genuine Truth calculated as the root-mean-square deviation (RMSD) between respective numerical experiments.

| Configuration | Phosphate RMSD | Phytoplankton RMSD |
|---|---|---|
| MIX+ | 0.16 $mmolP/m^3$ | 0.033 $mmolN/m^3$ |
| TUNE | 0.1 $mmolP/m^3$ | 0.021 $mmolN/m^3$ |

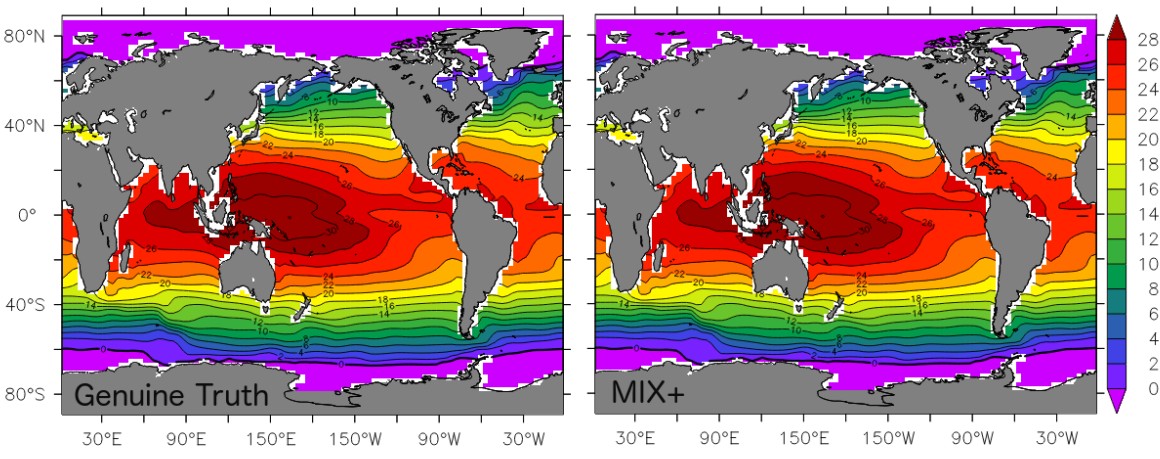

**Figure 1.** Simulated sea surface temperature in units $^\circ$C for the historical simulations (cf., Section 2.2.2). The left and right panel refer to results from the model configurations Genuine Truth and MIX+, respectively. MIX+ features an increased vertical background diffusivity relative to the Genuine Truth (cf., Table 3).





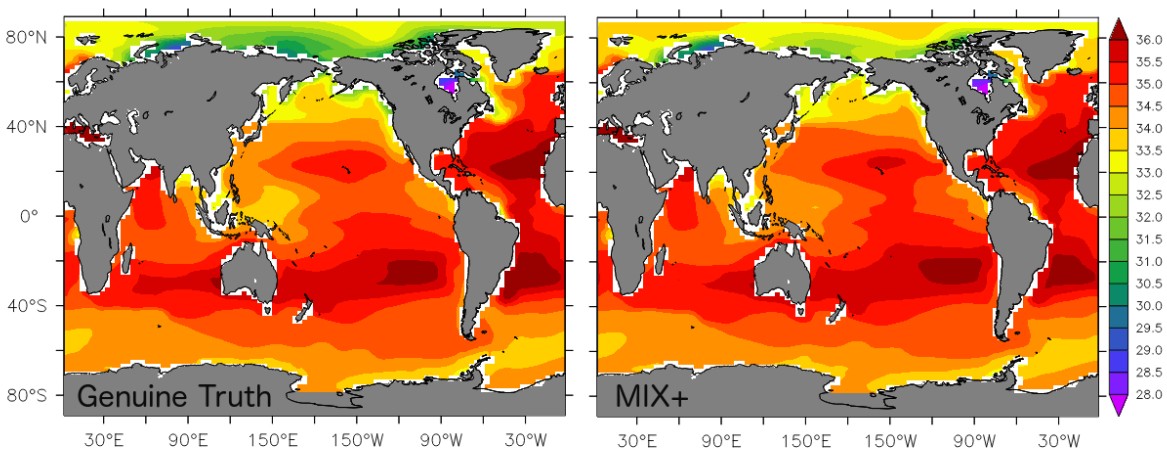

**Figure 2.** Simulated sea surface salinity in units PSU for the historical simulations (cf., Section 2.2.2). The left and right panel refer to results from the model configurations Genuine Truth and MIX+, respectively. MIX+ features an increased vertical background diffusivity relative to the Genuine Truth (cf., Table 3).

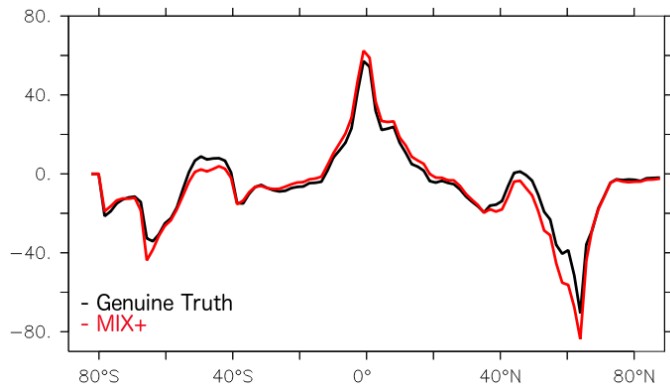

**Figure 3.** Simulated, zonally-averaged net air-sea heat fluxes in units $W\,m^{-2}$ for the historical simulations (cf., Section 2.2.2). Positive numbers denote ocean warming. The black and red line refer to results from the model configurations Genuine Truth and MIX+, respectively.





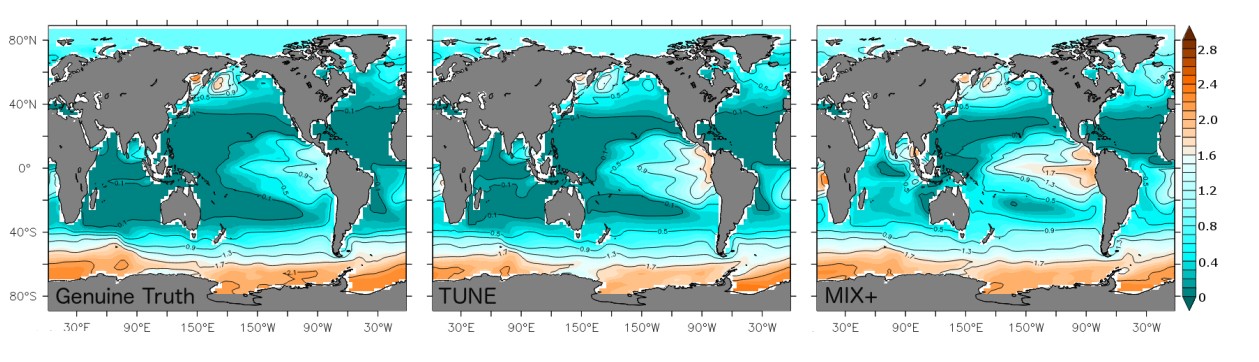

**Figure 4.** Simulated surface phosphate concentrations in units $mmolP/m^3$ for the historical simulations (cf., Section 2.2.2). The left, middle and right panel refer to results from the model configurations Genuine Truth, TUNE and MIX+, respectively. MIX+ and TUNE feature increased vertical background diffusivity relative to the Genuine Truth. In addition, TUNE features retuned biogeochemical model parameters (cf., Table 3).

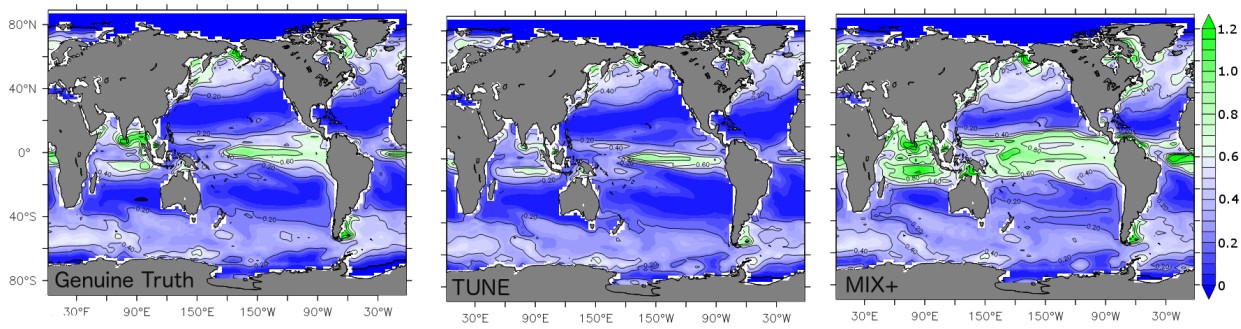

**Figure 5.** Simulated phytoplankton concentrations in units $mmolN/m^3$ for the historical simulations (cf., Section 2.2.2). The left, middle and right panel refer to results from the model configurations Genuine Truth, TUNE and MIX+, respectively. MIX+ and TUNE feature increased vertical background diffusivity relative to the Genuine Truth. In addition, TUNE features retuned biogeochemical model parameters (cf., Table 3).



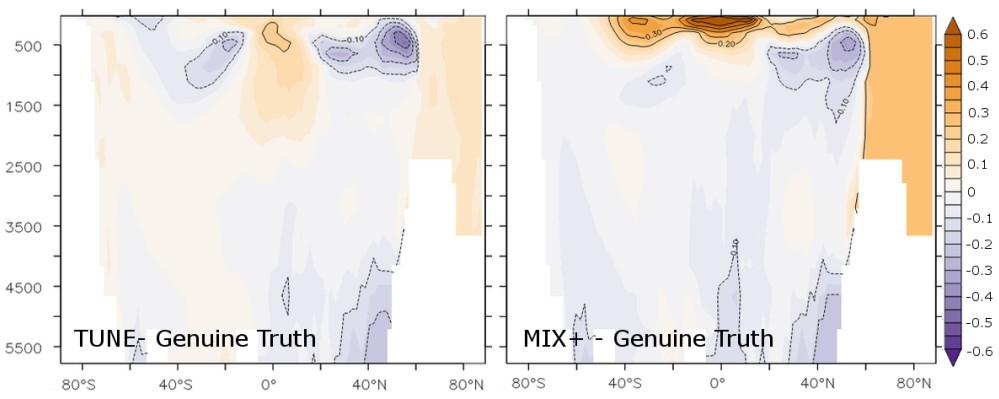

**Figure 6.** Differences between simulated meridionally-averaged phosphate concentrations in units $mmol\,P/m^3$ for the historical simulations (cf., Section 2.2.2). The left and right panel refer to TUNE minus Genuine Truth and MIX+ mins Genuine Truth, respectively.

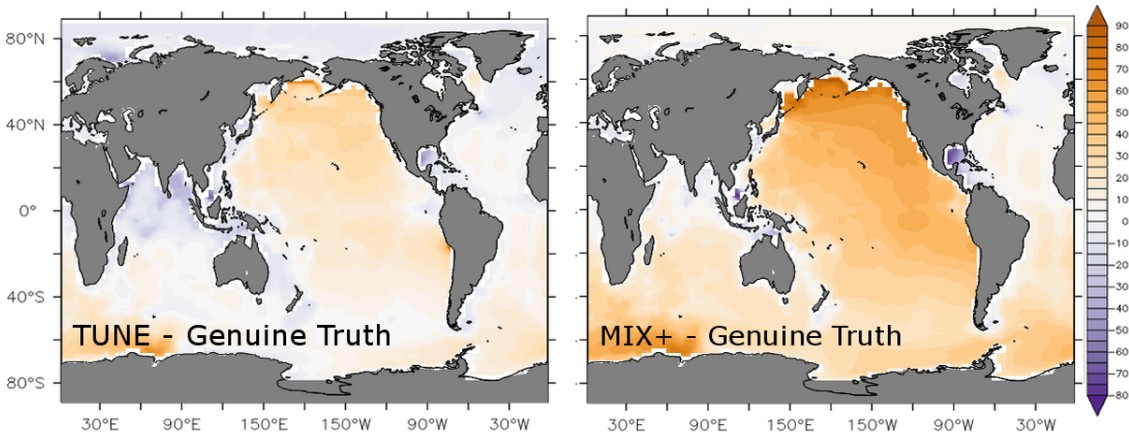

**Figure 7.** Differences between simulated depth-averaged oxygen concentrations in units $mmol\,O_2/m^3$ for the historical simulations (cf., Section 2.2.2). The left and right panel refer to TUNE minus Genuine Truth and MIX+ mins Genuine Truth, respectively.





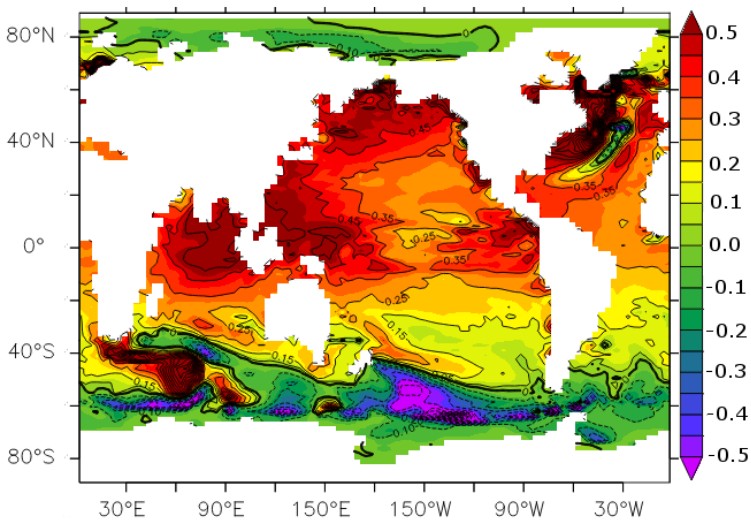

**Figure 8.** Comparison of sea surface temperature warming in response to RCP8.5 emissions. The contours (both colored and labeled) denote the differences in simulated sea surface temperature anomalies (year 2100 minus year 1850) in the projection based on the Genuine Truth setup relative to TUNE in units $K$. Positive numbers indicate regions where the Genuine Truth warmed more than TUNE in the year 2100.

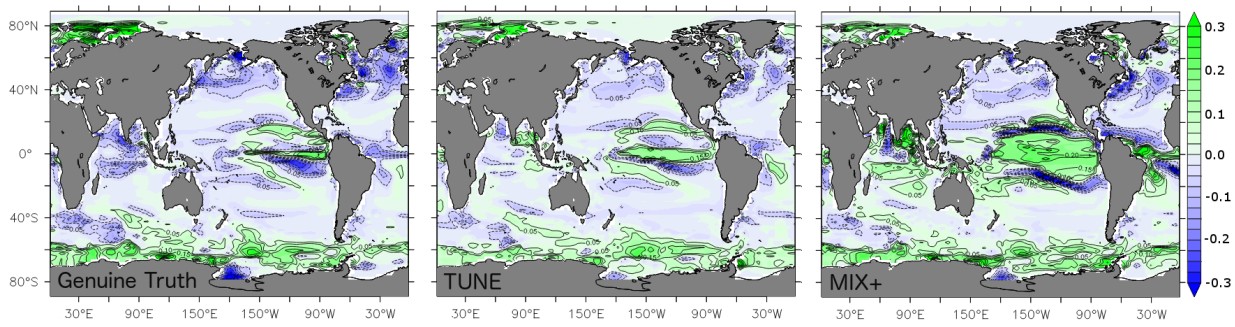

**Figure 9.** Simulated changes in surface phytoplankton concentrations as a consequence of rising $CO_2$ concentrations (emission scenario RCP 8.5) calculated as the annual mean concentration difference between the years 2100 - 1850 [in units $mmolN/m^3$].



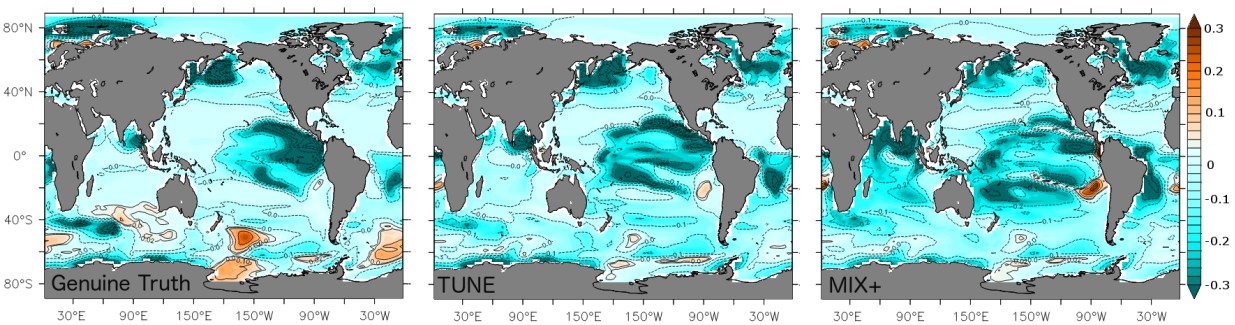

**Figure 10.** Simulated changes in surface phosphate concentrations as a consequence of rising $CO_2$ concentrations (emission scenario RCP 8.5) calculated as the annual mean concentration difference between the years 2100 - 1850 [in units $mmolP/m^3$].

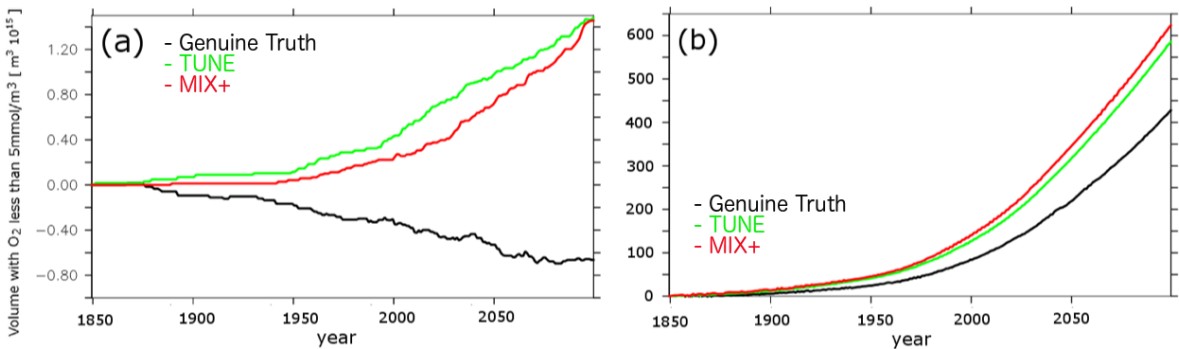

**Figure 11.** (a) Simulated temporal evolution of the volume of global suboxic waters (emission scenario RCP 8.5). (b) Differences in projected anomalous total carbon content of the ocean in response to rising CO2 concentrations (emission scenario RCP 8.5) [in units $GtC$].