# Peer review of "Reciprocal bias compensation and ensuing uncertainties in model-based climate projections: pelagic biogeochemistry versus ocean mixing."

_Biogeosciences, 2018_

## Referee Comment (RC1) · Omta (Referee) · 7 Jan 2019

General comments

A key problem in predicting the rate of carbon uptake by the oceans is that the uncertainty in the diapycnal diffusion. The authors show that this problem is not alleviated by optimizing an ocean model against the historical and current (biogeochemical) observations: the past and current state of the ocean can be described with very different parameter sets, which give dramatically different predictions for the future. Overall, this

is a decent piece of work that I recommend for publication after minor revisions.

Specific comment

Although I think this is a decent piece of work, the Conclusions section leaves me rather dissatisfied. In particular, I find the conclusion "that an improved understanding of vertical diapycnal mixing in Earth System Models alleviates the risk of reciprocal bias compensation by (wrongly) tweaking biogeochemical modules to a deficient physics" weak. It is of course great to diagnose a problem, but it would be helpful to have some more specific pointers to how future research could work toward a possible solution. In other words: how can we get to an improved understanding of vertical diapycnal mixing?

Technical comments

throughout: "reziprocal" -> "reciprocal", "brakes" -> "breaks"

p. 1, l. 12: "Metrics like total..." -> "Metrics such as total..."

p. 2, l. 15/16: "...of biogeochemical relevance), are contain various sources for uncertainties." -> "...of biogeochemical relevance), contain various sources of uncertainties."

p. 2, l. 18: "The reason being that diapycnal..." -> "The reason is that diapycnal..."

p. 3, l. 3/4: "Thus, typically, diapycnal mixing of a specific ocean model must have a profound impact on the respective biogeochemical component and it's parameter settings." -> "Therefore, diapycnal mixing of a specific ocean model typically has a strong impact on the respective biogeochemical component and its parameter settings."

p. 4, l. 12: "...primitive-equation model Pacanowski (2010)." -> "...primitive-equation model (Pacanowski, 2010)."

p. 5, l. 4: "Both, phytoplankton..." -> "Both phytoplnakton..."

p. 5, l. 5: "...remineralization rate determines the..." -> "...remineralization rate, determines the..."

p. 5, l. 8: "...mW denotes depth dependent detritus sinking speed..." -> "...mW is the derivative of wD with respect to depth..."

p. 6, l. 6: "...substantially different to the Genuine Truth." -> "...substantially different from the Genuine Truth."

p. 6, l. 28: "...they are representative for a pre-industrial world." -> "...they are representative of the pre-industrial world."

p. 7, l. 6: "In the following Section 3.1 we present..." -> "In the following Section 3.1, we present..."

p. 7, l. 7/8: "In Section 3.2 we present..." -> "In Section 3.2, we present..."

p. 8, l. 6: "...inline" -> "...in line..."

p. 8, l. 11-14: "The results under historical forcing are as follows: In terms of physical ocean circulation TUNE and MIX+ are almost identical (not shown). The main reason for this insensitivity is that the atmospheric CO2 concentrations are prescribed to the same preindustrial levels in both of the historical solutions. Hence the feedback from changed biogeochemistry via oceanic carbon uptake to atmospheric CO2 and associated changes in air-sea heat fluxes is excluded." -> "Under historical forcing, the physical ocean circulation is almost identical in TUNE and MIX+ (not shown). The main reason for this insensitivity is that the atmospheric CO2 concentrations are prescribed to the same preindustrial levels in both of the historical solutions, which means that the feedback from changed biogeochemistry via oceanic carbon uptake to atmospheric CO2 and associated changes in air-sea heat fluxes is excluded."

p. 9., l. 12: "...warms quicker than TUNE..." -> "...warms more quickly than TUNE..."

p. 9, l. 23: "Projections of phytoplankton are of interest also because phytoplankton forms..." -> "Projections of phytoplankton are of interest, because phytoplankton

forms..."

p. 9, l. 26: "...vertical mixing of nutrient replete waters..." -> "...vertical mixing of nutrient-replete waters..."

p. 10, l. 4: "Projections of phosphate are of interest also because it is an essential nutrient..." -> "Projections of phosphate are of interest, because phosphate is an essential nutrient..."

p. 10, l. 6: "...models phosphate..." -> "...models, phosphate..."

p. 10, l. 11" "...show alternating pattern of increasing..." -> "...show alternating patterns of increasing..."

p. 10, l. 18/19: "We speculate that, the absence of patches..." -> "We speculate that the absence of patches..."

p. 10, l. 28: "Projections of the oceanic carbon content are of interest, because the oceans currently take up..."

p. 10, l. 29: "To this end changes in..." -> "Therefore, changes in..."

p. 11, l. 1-3: "...oceanic carbon content: Historically, the ocean simulated carbon content of the ocean between the simulations varies by less then 0.5% between the simulations." -> "...oceanic carbon content. In the Historical Model Solutions, the simulated carbon content of the ocean varies by less than 0.5% between the simulations."

p. 11, l. 7/8: "Projections of suboxic volume are of interest also because suboxia trigger denitrification and thus reduce the global availability of fixed nitrogen which is an essential nutrient for all phytoplankton other than diazotrophs." -> "Projections of suboxic volume are of interest, because suboxia triggers denitrification and thus reduces the global availability of fixed nitrogen, an essential nutrient for all phytoplankton other than diazotrophs."

p. 11, l. 31 - p. 12, l. 1: "...that since, EMICs and...." -> "...that since EMICs and..."

p. 12, l. 2: "...kernels, that our..." -> "...kernels, our..."

p. 12, l. 3/4: "... are inline with our results. Still we have to stress that our mixing parameter settings can presumably be regarded..." -> "...are consistent with our results. Even so, we wish to stress that our mixing parameter settings should be regarded..."

p. 12, l. 20-24: "Hence, using a diffusivity that is averaged over time and space (as is inevitable in the current generation of models that apply a finite-differences discretization) but, at the same time, is fraught with high uncertainties. An additional source of uncertainty is implicit diffusion. As opposed to explicitly prescribed diffusion implicit diffusion is a spurious and hard-to-quantify artifact (cf., Getzlaff et al. (2012) ) of the underlying numerical advection algorithm." -> "Thus, using a diffusivity that is averaged over time and space (as is inevitable in the current generation of models that apply a finite-differences discretization) is fraught with uncertainties. An additional source of uncertainty is implicit diffusion, a spurious and hard-to-quantify artifact (cf., Getzlaff et al. (2012) ) of the underlying numerical advection algorithm."

p. 12, l. 28-31: "The visible effects on physical tracers when using ad-hoc measures, based on temperature and salinity, are rather weak. In terms of global sea surface temperature differences only 0.03K. In terms of meridional averaged heat fluxes the differences are below 5W/m2 form 50âŮęS to 50âŮęN. At high latitudes differences reach 25W/m2." -> "Using ad-hoc measures, based on temperature and salinity, has a rather weak impact on physical tracers: only 0.03K in terms of global sea surface temperature differences and in terms of meridional averaged heat fluxes, the differences are below 5W/m2 from 50âŮęS to 50âŮęN, reaching 25W/m2 at high latitudes."

p. 12, l. 33/34: "To rate this numbers, we put them into perspective with IPCC models:" -> "These numbers are broadly consistent with IPCC models:"

p. 13, l. 2-5: "The spread among models in the late twentieth century is with 21 Sv huge, and even exceeds the spread of the simulated max. meridional overturning in CMIP3 models. Thus, the difference between the lowest and highest projected max

overturning in CMIP5 models is even higher as the present day observational esti-
mate..." -> "The spread among models in the late twentieth century is with 21 Sv huge,
even exceeding the present day observational estimate..."

p. 13, l. 7: "...both, changed..." -> "...both changed..."

p. 13, l. 10: "E.g. Kriest et al. (2017) assumes..." -> "For example, Kriest et al. (2017)
assume..."

p. 13, l. 11: "...Illyina..." -> "...Ilyina..."

p. 13, l. 11: "Also,..." -> "Furthermore,..."

p. 13, l. 29: "...inline..." -> "...in line..."

p. 14, l. 1: "...wind forcing and Bryan (1987) stresses the importance for the simulated
large-scale meridional overturning..." -> "...wind forcing and thus the simulated large-
scale meridional overturning..."

p. 14, l. 6: "The results for suboxic volume is consistent with findings..." -> "The results
for suboxic volume are consistent with findings..."

p. 14, l. 14/15: "With this our results are inline with earlier findings..." -> "Our results
are consistent with earlier findings..."

p. 14, l. 23/24: "Also, dating ranges of CFCs are not suitable to resolve the dynamics
of the deep ocean which recently lead to the suggestion to introduce additionally 39Ar
(Ebser et al, 2018). The latter, promising approach is currently under investigation." ->
"Furthermore, dating ranges of CFCs are not suitable to resolve the dynamics of the
deep ocean which recently led to an investigation of the use of 39Ar as an additional
tracer (Ebser et al, 2018)."

p. 14, l. 25: "...an pressing..." -> "...a pressing..."

p. 15, l. 10: "For carbon the projections..." -> "For carbon, the projections..."

p .16, l. 9: "These results are inline with earlier findings..." -> "These results are consistent with earlier findings..."

p. 16, l. 17/18: "With these findings our study reminds on the need..." -> "With these findings, our study emphasizes the need..."

---

## Referee Comment (RC2) · Anonymous Referee #2 · 10 Jan 2019

The authors examine the effects of "reciprocal bias compensation", (compensating for errors in the physical model component of an earth system model with changes to the biogeochemical component) using a set of experiments with an earth system model of intermediate complexity. Overall, the manuscript makes an important point that is based on a simple experiment which is easy to follow. However at times it is difficult to follow the manuscript, some important information is missing, while other points are mentioned several times.

general comments:

At times, the manuscript seems very repetitive and I have tried to point out most of those instances in my specific comments below. This aspect mostly affects the second half of the manuscript; in fact, in the current version, the "Summary and Conclusions" section is followed by another "Conclusions" section. Section 6 repeats points that have been made in Section 5 which repeats many aspects described in Section 4. A thoughtful restructuring of the last sections would make the manuscript more concise and much easier to follow.

The phrase "reciprocal bias compensation" is used a lot in the manuscript but the actual bias is never examined. Beside 4 RMSD values in Table 4, differences between the configurations are not quantified. I would suggest to include bias values and more than just 2 variables in Table 4. It should include at least those variables examined in the manuscript (including physical variables) and could further include values for the projection into the future (4 columns: "MIX+ (historical)", "TUNE (historical)", "MIX+ (future)", "TUNE (future)"; variables appear in rows). Such a table would show the effects of reciprocal bias compensation on the different variables in one place, which would be very beneficial to the reader.

How were the parameters chosen for the spinup experiment used to determine the parameters for TUNE? I am not against a simple way to determine these parameter values but it would be good to know what ranges were considered and how the values were determined ... grid search, randomly chosen from an interval, latin hypercube sampling? More information is needed.

One interesting experiment would be to include the $\kappa_h$ (vertical diffusion) parameter in an updated TUNE experiment, just to see if in this very simple case (all error is one parameter) the original value of $\kappa_h$ could be recovered or if corrections to the biological parameters and thus reciprocal bias compensation would prevail. No further experiments (generating new projections into the future) would be necessary for this experiment.

specific comments:

p1 l7: It is not quite clear what "approach" is referring to.

p1 l14: "Vice versa" -> "Likewise"

p1 l17: remove "away"

p1 l19: "effectivity" -> "effectiveness"

p2 l1: "to" -> "with"

p2 l4: I would not call climatological products "observations"; "data" may be more suitable here.

p2 l6: It's not quite clear what "generic problems" refers to.

p2 l14: "movement" -> "transport"

p2 l15: "are contain various sources for uncertainties" -> "contain various sources of uncertainty"

p2 l18: "The reason ...": please simplify this sentence.

p2 l21: "pump of carbon" -> "carbon pump"

p3 l2: "when the biogeochemical model is unchanged." Is this different if the biogeochemical model is changed? I would remove this part of the sentence.

p3 l3: "must have a profound impact on the respective biogeochemical component and it's parameter settings" I would suggest to change this to: "has a profound impact on the respective biogeochemical component and should inform its parameter settings".

p3 l4: "The latter holds particularly as the development of Earth System Models is modular and module after module is coupled together." I am not sure what is meant here, rephrase?

p3 l13: "the value of diapycnal diffusivity, that is to be set in models" I'd suggest: "any

given value of diapycnal diffusivity that is used in a model"

p3 l14: "physics" -> "physical model component"

p3 l19: "based on" -> "using"

p3 l21: "biased ocean component" -> "biased physical ocean component"

p3 l25: "The model setups..." You could include the configuration names for each version in this sentence (include "MIX+" and "TUNE").

p4 l19: "dissolved inorganic carbon (DIN)": Shouldn't that be DIC?

p4 l20: "tuned tuned"

eq1: Typically, T and S denote temperature and salinity (and on page 5 "T" is used for temperature), please use different symbols here.

p4 l28: There is a "(3)" missing and \mu_D0 is called \mu_D in Table 1.

p4 l31: "iron mask" Surely not a face mask, what is this exactly?

p5 l1: "In one of our configurations we change..." I think it is more useful to specify the configuration or say "in our parameter tuning experiment we change" (this applies to all 3 instances of this sentence).

p5 l5: "depth" -> "depth range"

p5 l5: sometimes "nitrate" is used, sometimes "NO3"

p5 l6: "ends its route down to the sea-floor" -> "stops sinking"

p5 l8: w_D0 has already been described.

p5 l23: The reference to "ground truth" is not useful here and just confuses the reader.

p5 l26: "We define this model configuration as ground truth" this has now been mentioned multiple times.

p6 l5: "The leading thought behind these changes relative to MIX+ is to mimic the behavior of the Genuine Truth configuration even though the vertical background diffusion is substantially different to the Genuine Truth. Or, in other words, ..." This has been mentioned before and can be simplified

p6 l9: Point (1) is not part of the procedure and should appear in section 2.1 to motivate the choice of parameters there.

p6 l14: How was chlorophyll a converted via Redfield?

p6 l14: "surface chlorophyll a and oceanic phosphate": where the observations distributed evenly across the globe? Does "oceanic phosphate" imply surface values? More information is needed on how the best fit was determined.

p6 l19: It would be nice to actually list the bias alongside the RMSD in table 4.

p6 l30: It is not clear to me what the difference between the spinup and the drift runs are if both use pre-industrial atmospheric CO2.

p7 l3: This summary is repeated again in the next paragraph.

p7 l15: "Mix+" -> "MIX+"

p8 l1: "antagonistic": do you mean "twofold"? Please rephrase.

p8 l5 (and elsewhere): "inline" -> "in line" (or reword using for example "agree with")

p8 l24: "bias" but those are RMSD values, please report the bias as well.

p9 l4: It would be good to mention this feedback mechanism when the model is introduced.

p9 l7: In this paragraph the language makes it seem like Genuine Truth is evaluated with respect to the TUNE configuration (same comment applies to Fig 8), it is better to switch it around, that is: "TUNE shows ... relative to Genuine Truth."

p9 l23: Remove "also" from this sentence and following introductory sentences for

other subsections.

p10 l6: Doesn't "base currency" imply that variables like phytoplankton is measured in units of phosphate concentration?

p11 l1: Add "(b)" to figure reference (or better, switch the panel order and add "(a)").

p12 l12: The notion of "slightly changed physics" does not seem to agree with "we have to stress that our mixing parameter settings can presumably be regarded as extreme cases" just a few lines above. I would suggest to reword, emphasizing that just one physical parameter was modified.

p12 l26: "The latter change is motivated...": It is difficult to follow this and the next sentence. Please rephrase or delete.

p13 l3: "is with 21 Sv huge": is this the ensemble spread from the Reintges et al. (2017) report? If so, make this explicit; if not, please rephrase, it is difficult for the reader to follow the train of thought here.

p13 l8: "have been changed by 12.5, ≈200 and 115% respectively": use "%" for all numbers!

p13 l26: "In terms of the uncertainties in projected sea surface temperatures imposed by ocean mixing are locally substantial:" Something is missing in this sentence

p14 l17: "suited" -> "suitable"

Fig 2: combine with Fig 1. I would be good to show the difference between the two panels.

Fig 4, 5 and others: It would be very useful to have difference maps. If there is not enough room for extra panels I would suggest to show the regular field for Genuine Truth but difference fields for the two other configurations.

Fig 6: A label for the y-axis is missing and the background should have a color that is

not in the the color map (grey?).

Fig 6: caption says "mins"

Fig 7: Here, one the other hand, it may be useful to show the oxygen conc of the Genuine Trurth in a new first panel.

Fig 8: Differences in anomalies are not that easy to interpret. I suppose the temperature in 1850 is very similar in both simulations, so the difference in anomalies is basically a difference in temperature? If that is the case, mention this. Furthermore, based on the description, why is Genuine Truth - TUNE shown and not the other way around, like in all other plots? I would suggest to use three panels again: (1) anomaly for Genuine Truth, (2) difference TUNE - Genuine Truth (3) difference MIX+ - Genuine Truth.

Fig 11: add description and units to panel (b). It would makes sense to switch panels, so that they appear in the same order as in the manuscript text.

Table 3: combine with Table 1.

---

## Author Comment (AC2) · 31 Jan 2019

**Reviewer #2 (anonymous)**

R: The authors examine the effects of "reciprocal bias compensation", (compensating for errors in the physical model component of an earth system model with changes to the biogeochemical component) using a set of experiments with an earth system model of intermediate complexity. Overall, the manuscript makes an important point that is based on a simple experiment which is easy to follow. However at times it is difficult to follow the manuscript, some important information is missing, while other points are mentioned several times.

*A: The reviewer made constructive suggestions targeted at making the paper less repetitive and, further, he asks for (some) missing information. We will follow his suggestions in a revised version of the manuscript (which we deem as being, due to the constructive nature of the reviewer's comment, rather straight forward). We thank the reviewer for his/her time and effort!*

R: general comments:
At times, the manuscript seems very repetitive and I have tried to point out most of those instances in my specific comments below. This aspect mostly affects the second half of the manuscript; in fact, in the current version, the "Summary and Conclusions" section is followed by another "Conclusions" section. Section 6 repeats points that have been made in Section 5 which repeats many aspects described in Section 4. A thoughtful restructuring of the last sections would make the manuscript more concise and much easier to follow.

*A: We agree with the reviewer and will go through the text and exclude repetitions.*

The phrase "reciprocal bias compensation" is used a lot in the manuscript but the actual bias is never examined. Beside 4 RMSD values in Table 4, differences between the configurations are not quantified. I would suggest to include bias values and more than just 2 variables in Table 4. It should include at least those variables examined in the manuscript (including physical variables) and could further include values for the projection into the future (4 columns: "MIX+ (historical)", "TUNE (historical)", "MIX+ (future)", "TUNE (future)"; variables appear in rows). Such a table would show the effects of reciprocal bias compensation on the different variables in one place, which would be very beneficial to the reader.

*A: This is a very good suggestion and we will extend the respective Table as suggested.*

How were the parameters chosen for the spinup experiment used to determine the parameters for TUNE? I am not against a simple way to determine these parameter values but it would be good to know what ranges were considered and how the values were determined ... grid search, randomly chosen from an interval, latin hypercube sampling? More information is needed.

*A: Agreed! We will add the respective information.*

One interesting experiment would be to include the nkappa_h (vertical diffusion) parameter in an updated TUNE experiment, just to see if in this very simple case (all error is one parameter) the original value of nkappa_h could be recovered or if corrections to the biological parameters and thus reciprocal bias compensation would prevail. No further experiments (generating new projections into the future) would be necessary for this experiment.

*A: Agreed! We would, however, like to combine these types of experiments with a new cost function which gives some measure of the realism of the effective mixing. Our current research indicates that the saturation state of noble gases such as argon may be suited to provide such a measure. We will add a respective outlook to the discussion section of our revised manuscript.*

R: specific comments
*A: We thank the reviewer for investing all this time! All the comments are very constructive and helpful! We will follow all suggestions made by the reviewer.*

---

## Author Response (AR1)

**Reviewer #1 (Omta)**

R: General comments
A key problem in predicting the rate of carbon uptake by the oceans is the uncertainty
in the diapycnal diffusion. The authors show that this problem is not alleviated by
optimizing an ocean model against the historical and current (biogeochemical) observations:
the past and current state of the ocean can be described with very different
parameter sets, which give dramatically different predictions for the future. Overall, this is a decent piece of work
that I recommend for publication after minor revisions.

*A: We thank A.W. Omta for his time and effort! We found his comments very helpful and considered all of them in the revised version of our manuscript. Detailed answers as well as a marked-up manuscript version are provided below.*

R: Specific comment
Although I think this is a decent piece of work, the Conclusions section leaves me rather dissatisfied. In particular, I find the conclusion "that an improved understanding of vertical diapycnal mixing in Earth System Models alleviates the risk of reciprocal bias compensation by (wrongly) tweaking biogeochemical modules to a deficient physics" weak. It is of course great to diagnose a problem, but it would be helpful to have some more specific pointers to how future research could work toward a possible solution. In other words: how can we get to an improved understanding of vertical diapycnal mixing?

*A: Our ongoing work indicates that the saturation state of noble gases such as argon may be applied to constrain diapycnal mixing in ocean models. We added this outlook to the Discussion part of the revised manuscript (page 15, line 15ff).*

R: Technical comments

*A: We thank A.W. Omta for the extensive list of corrections. We, gratefully, considered all of them.*

R: throughout: "reziprocal" -> "reciprocal", "brakes" -> "breaks"
**A: Changed (e.g., page 7, line 23 and 24; page 11, line 23)**
R: p. 1, l. 12: "Metrics like total..." -> "Metrics such as total..."
**A: Changed (page 1, line 12)**
R: p. 2, l. 15/16: "...of biogeochemical relevance), are contain various sources for uncertainties."
-> "...of biogeochemical relevance), contain various sources of uncertainties."
**A: Changed (page 2, line 15)**
R: p. 2, l. 18: "The reason being that diapycnal..." -> "The reason is that diapycnal..."
**A: Changed (page 2, line 18)**
R: p. 3, l. 3/4: "Thus, typically, diapycnal mixing of a specific ocean model must have a
profound impact on the respective biogeochemical component and it's parameter settings."
-> "Therefore, diapycnal mixing of a specific ocean model typically has a strong
impact on the respective biogeochemical component and its parameter settings."
**A: Reformulated (page 3, line 3f)**
R: p. 4, l. 12: "...primitive-equation model Pacanowski (2010)." -> "...primitive-equation
model (Pacanowski, 2010)."
**A: Changed (page 4, line 15)**
R: p. 5, l. 4: "Both, phytoplankton..." -> "Both phytoplnakton..."
**A: Changed (page 5, line 11)**
R: p. 5, l. 5: "...remineralization rate determines the..." -> "...remineralization rate, determines the..."
**A: Comma added (page 5, line 12)**
R: p. 5, l. 8: "...mW denotes depth dependent detritus sinking speed..." -> "...mW is the
derivative of wD with respect to depth..."
**A: Reformulated (page 5, line 15)**
R: p. 6, l. 6: "...substantially different to the Genuine Truth." -> "...substantially different
from the Genuine Truth."
**A: Changed (page 6, line 13)**
R: p. 6, l. 28: "...they are representative for a pre-industrial world." -> "...they are representative
of the pre-industrial world."
**A: Changed (page 7, line 6)**
R: p. 7, l. 6: "In the following Section 3.1 we present..." -> "In the following Section 3.1,
we present..."
**A: Changed (page 7, line 22)**
R: p. 7, l. 7/8: "In Section 3.2 we present..." -> "In Section 3.2, we present..."
**A: Changed (page 7, line 23)**
R: p. 8, l. 6: "...inline" -> "...in line..."
**A: Changed (page 8, line 24)**
R: p. 8, l. 11-14: "The results under historical forcing are as follows: In terms of physical

ocean circulation TUNE and MIX+ are almost identical (not shown). The main reason for this insensitivity is that the atmospheric CO2 concentrations are prescribed to the same preindustrial levels in both of the historical solutions. Hence the feedback from changed biogeochemistry via oceanic carbon uptake to atmospheric CO2 and associated changes in air-sea heat fluxes is excluded." -> "Under historical forcing, the physical ocean circulation is almost identical in TUNE and MIX+ (not shown). The main reason for this insensitivity is that the atmospheric CO2 concentrations are prescribed to the same preindustrial levels in both of the historical solutions, which means that the feedback from changed biogeochemistry via oceanic carbon uptake to atmospheric CO2 and associated changes in air-sea heat fluxes is excluded."

**A: Changed (page 8, line 29ff)**

R: p. 9., l. 12: "...warms quicker than TUNE..." -> "...warms more quickly than TUNE..."

**A: Changed (page 10, line 2)**

R: p. 9, l. 23: "Projections of phytoplankton are of interest also because phytoplankton forms..."

**A: Changed (page 10, line 13)**

R: p. 9, l. 26: "...vertical mixing of nutrient replete waters..." -> "...vertical mixing of nutrient-replete waters..."

**A: Changed (page 10, line 16)**

R: p. 10, l. 4: "Projections of phosphate are of interest also because it is an essential nutrient..." -> "Projections of phosphate are of interest, because phosphate is an essential nutrient..."

**A: Changed (page 10, line 27)**

R: p. 10, l. 6: "...models phosphate..." -> "...models, phosphate..."

**A: Comma added (page 10, line 29)**

R: p. 10, l. 11" "...show alternating pattern of increasing..." -> "...show alternating patterns of increasing..."

**A: Changed (page 11, line 1)**

R: p. 10, l. 18/19: "We speculate that, the absence of patches..." -> "We speculate that the absence of patches..."

**A: Changed (page 11, line 9)**

R: p. 10, l. 28: "Projections of the oceanic carbon content are of interest, because the oceans currently take up..."

**A: Changed (page 11, line 18)**

R: p. 10, l. 29: "To this end changes in..." -> "Therefore, changes in..."

**A: Changed (page 11, line 19)**

R: p. 11, l. 1-3: "...oceanic carbon content: Historically, the ocean simulated carbon content of the ocean between the simulations varies by less then 0.5% between the simulations." -> "...oceanic carbon content. In the Historical Model Solutions, the simulated carbon content of the ocean varies by less than 0.5% between the simulations."

**A: Changed (page 11, line 24)**

R: p. 11, l. 7/8: "Projections of suboxic volume are of interest also because suboxia trigger denitrification and thus reduce the global availability of fixed nitrogen which is an essential nutrient for all phytoplankton other than diazotrophs." -> "Projections of suboxic volume are of interest, because suboxia triggers denitrification and thus reduces the global availability of fixed nitrogen, an essential nutrient for all phytoplankton other than diazotrophs."

**A: Changed (page 11, line 29ff)**

R: p. 11, l. 31 - p. 12, l. 1: "...that since, EMICs and...." -> "...that since EMICs and..." forms..."

**A: Changed (page 12, line 21)**

R: p. 12, l. 2: "...kernels, that our..." -> "...kernels, our..."

**A: Changed (page 12, line 22)**

R: p. 12, l. 3/4: "... are inline with our results. Still we have to stress that our mixing parameter settings can presumably be regarded..." -> "...are consistent with our results. Even so, we wish to stress that our mixing parameter settings should be regarded..."

**A: Changed (page 12, line 23)**

R: p. 12, l. 20-24: "Hence, using a diffusivity that is averaged over time and space (as is inevitable in the current generation of models that apply a finite-differences discretization) but, at the same time, is fraught with high uncertainties. An additional source of uncertainty is implicit diffusion. As opposed to explicitly prescribed diffusion implicit diffusion is a spurious and hard-to-quantify artifact (cf., Getzlaff et al. (2012) ) of the underlying numerical advection algorithm." -> "Thus, using a diffusivity that is averaged over time and space (as is inevitable in the current generation of models that apply a finite-differences discretization) is fraught with uncertainties. An additional source of uncertainty is implicit diffusion, a spurious and hard-to-quantify artifact (cf., Getzlaff et al. (2012) ) of the underlying numerical advection algorithm."

**A: Changed (page 13, line 11ff)**

R: p. 12, l. 28-31: "The visible effects on physical tracers when using ad-hoc measures, based on temperature and salinity, are rather weak. In terms of global sea surface

temperature differences only 0.03K. In terms of meridional averaged heat fluxes the differences are below 5W/m2 form 50°S to 50°N. At high latitudes differences reach 25W/m2." -> "Using ad-hoc measures, based on temperature and salinity, has a rather weak impact on physical tracers: only 0.03K in terms of global sea surface temperature differences and in terms of meridional averaged heat fluxes, the differences are below 5W/m2 from 50°S to 50°N, reaching 25W/m2 at high latitudes."
**A: Reformulated (page 13, line 18ff)**
R: p. 12, l. 33/34: "To rate this numbers, we put them into perspective with IPCC models:" -> "These numbers are broadly consistent with IPCC models:"
**A: Changed (page 13, line 23)**
R: p. 13, l. 2-5: "The spread among models in the late twentieth century is with 21 Sv huge, and even exceeds the spread of the simulated max. meridional overturning in CMIP3 models. Thus, the difference between the lowest and highest projected max overturning in CMIP5 models is even higher as the present day observational estimate..." -> "The spread among models in the late twentieth century is with 21 Sv huge, even exceeding the present day observational estimate..."
**A: Reformulated (page 13, line 26ff)**
R: p. 13, l. 7: "...both, changed..." -> "...both changed..."
**A: Changed (page 13, line 31)**
R: p. 13, l. 10: "E.g. Kriest et al. (2017) assumes..." -> "For example, Kriest et al. (2017) assume..."
**A: Changed (page 13, line 33)**
R: p. 13, l. 11: "...Illyina..." -> "...Ilyina..."
**A: Changed (page 14, line 2)**
R: p. 13, l. 11: "Also,..." -> "Furthermore,..."
R: p. 13, l. 29: "...inline..." -> "...in line..."
**A: Changed (page 14, line 20)**
R: p. 14, l. 1: "...wind forcing and Bryan (1987) stresses the importance for the simulated large-scale meridional overturning..." -> "...wind forcing and thus the simulated largescale meridional overturning..."
**A: Changed (page 14, line 22)**
R: p. 14, l. 6: "The results for suboxic volume is consistent with findings..." -> "The results for suboxic volume are consistent with findings..."
**A: Changed (page 14, line 26/27)**
R: p. 14, l. 14/15: "With this our results are inline with earlier findings..." -> "Our results are consistent with earlier findings..."
**A: Changed (page 15, line 3)**
R: p. 14, l. 23/24: "Also, dating ranges of CFCs are not suitable to resolve the dynamics of the deep ocean which recently lead to the suggestion to introduce additionally 39Ar (Ebser et al, 2018). The latter, promising approach is currently under investigation." -> "Furthermore, dating ranges of CFCs are not suitable to resolve the dynamics of the deep ocean which recently led to an investigation of the use of 39Ar as an additional tracer (Ebser et al, 2018)."
**A: Changed (page 15, line 11ff)**
R: p. 14, l. 25: "...an pressing..." -> "...a pressing..."
**A: Changed (page 15, line 14)**
R: p. 15, l. 10: "For carbon the projections..." -> "For carbon, the projections..."
**A: The sentence was removed.**
R: p .16, l. 9: "These results are inline with earlier findings..." -> "These results are consistent with earlier findings..."
**A: Changed (page 16, line 7)**
R: p. 16, l. 17/18: "With these findings our study reminds on the need..." -> "With these findings, our study emphasizes the need..."
**A: Comma added (page 16, line 16)**

**Reviewer #2 (anonymous)**
*We thank the reviewer for his/her time and effort and very helpful comments! Detailed answers to the comments as well as a marked-up manuscript version are provided below.*

R: The authors examine the effects of "reciprocal bias compensation", (compensating for errors in the physical model component of an earth system model with changes to the biogeochemical component) using a set of experiments with an earth system model of intermediate complexity. Overall, the manuscript makes an important point that is based on a simple experiment which is easy to follow. However at times it is difficult to follow the manuscript, some important information is missing, while other points are mentioned several times.

*A: The reviewer made constructive suggestions targeted at making the paper less repetitive and, further, he asks for (some) missing information. We followed his/her suggestions in a revised version of the manuscript (e.g., we removed the repetitive Section 6 and extended the description of the experimental design on page 6). Details regarding these changes are provided below as answers to the more specific comments.*

R: general comments:
At times, the manuscript seems very repetitive and I have tried to point out most of those instances in my specific comments below. This aspect mostly affects the second half of the manuscript; in fact, in the current version, the "Summary and Conclusions" section is followed by another "Conclusions" section. Section 6 repeats points that have been made in Section 5 which repeats many aspects described in Section 4. A thoughtful restructuring of the last sections would make the manuscript more concise and much easier to follow.

*A: We thank the reviewer for pointing this out and went carefully through the text to exclude repetitions. Specifically, we deleted Section 6 in the revised version of the manuscript and summarized all conclusions in Section 5.*

The phrase "reciprocal bias compensation" is used a lot in the manuscript but the actual bias is never examined. Beside 4 RMSD values in Table 4, differences between the configurations are not quantified. I would suggest to include bias values and more than just 2 variables in Table 4. It should include at least those variables examined in the manuscript (including physical variables) and could further include values for the projection into the future (4 columns: "MIX+ (historical)", "TUNE (historical)", "MIX+ (future)", "TUNE (future)"; variables appear in rows). Such a table would show the effects of reciprocal bias compensation on the different variables in one place, which would be very beneficial to the reader.

*A: This is a very good suggestion and we extended the respective Table 2 as suggested. Biases have been added to the text (e.g., page 9, line 1,2 and 7). Also we highlight now clearly the fact that the different model sensitivities already impact the drift runs (page 7, line 15ff).*

How were the parameters chosen for the spinup experiment used to determine the parameters for TUNE? I am not against a simple way to determine these parameter values but it would be good to know what ranges were considered and how the values were determined ... grid search, randomly chosen from an interval, latin hypercube sampling? More information is needed.

*A: Agreed! We added the respective information (page 6, line 29ff).*

One interesting experiment would be to include the nkappa_h (vertical diffusion) parameter in an updated TUNE experiment, just to see if in this very simple case (all error is one parameter) the original value of nkappa_h could be recovered or if corrections to the biological parameters and thus reciprocal bias compensation would prevail. No further experiments (generating new projections into the future) would be necessary for this experiment.

*A: Agreed! We would, however, like to combine these types of experiments with a new cost function which gives some measure of the realism of the effective mixing. Our current research indicates that the saturation state of noble gases such as argon may be suited to provide such a measure. We added a respective outlook to the discussion section of our revised manuscript (page 15, line 15ff).*

specific comments:
**A: We thank the reviewer for extensively going through the text and considered all comments. Detailed answers are given below.**

p1 l7: It is not quite clear what "approach" is referring to.
**A: We added the respective information (page 13, line 17).**
p1 l14: "Vice versa" -> "Likewise"  changed "Related to the latter"
**A: Reformulated (page 1, line 14)**
p1 l17: remove "away" sequestration from the atmosphere
**A: Deleted (page 1, line 18)**
p1 l19: "effectivity" -> "effectiveness"
**A: Changed (page 1, line 19)**
p2 l1: "to" -> "with"
**A: Changed (page 2, line 1)**
p2 l4: I would not call climatological products "observations"; "data" may be more
suitable here.
**A: Changed (page 2, line 4)**
p2 l6: It's not quite clear what "generic problems" refers to.
**A: Changed (page 2, line 6/7)**

p2 l14: "movement" -> "transport"
**A: Changed (page 2, line 14)**
p2 l15: "are contain various sources for uncertainties"
**A: Reformulated (page 2, line 15/16)**
p2 l18: "The reason ...": please simplify this sentence.
**A: Reformulated (page 2, line 18)**
p2 l21: "pump of carbon" -> "carbon pump"
**A: Changed (page 2, line 21)**
p3 l2: "when the biogeochemical model is unchanged." Is this different if the biogeo-
chemical model is changed? I would remove this part of the sentence.
**A: Done**
p3 l3: "must have a profound impact on the respective biogeochemical component and
it's parameter settings" I would suggest to change this to: "has a profound impact on
the respective biogeochemical component and should inform its parameter settings".
**A: Reformulated (page 3, line 3/4)**
p3 l4: "The latter holds particularly as the development of Earth System Models is
modular and module after module is coupled together." I am not sure what is meant
here, rephrase?
**A: Reformulated (page 3, line 5)**
p3 l13: "the value of diapycnal diffusivity, that is to be set in models" I'd suggest: "any
given value of diapycnal diffusivity that is used in a model"
**A: Reformulated (page 3, line 14)**
p3 l14: "physics" -> "physical model component"
**A: Changed (page 3, line 22)**
p3 l19: "based on" -> "using"
**A: Changed (page 3, line 20)**
p3 l21: "biased ocean component" -> "biased physical ocean component"
**A: Changed (page 3, line 22)**
p3 l25: "The model setups..." You could include the configuration names for each version in this sentence
(include "MIX+" and "TUNE").
**A; These names were not yet introduced.**
p4 l19: "dissolved inorganic carbon (DIN)": Shouldn't that be DIC?
**A: Thanks + changed (page 4, line 22)**
p4 l20: "tuned tuned"
**A: Changed (page 4, line 23)**
eq1: Typically, T and S denote temperature and salinity (and on page 5 "T" is used for
temperature), please use different symbols here. *-> changed to DTR and SRC*
**A: True – we changed "T" to "DTR" and "S" to "SRC" (page 4, line 25-27)**
p4 l28: There is a "(3)" missing and \mu_D0 is called \mu_D in Table 1.
**A: Thanks + we changed that in Tab.1**
p4 l31: "iron mask" Surely not a face mask, what is this exactly? *-> clarified*
**A: True + we added a short explanation (page 5, line 2).**
p5 l1: "In one of our configurations we change..." I think it is more useful to specify the
configuration or say "in our parameter tuning experiment we change" (this applies to all
3 instances of this sentence).
**A: Changed (page 5, line 8)**
p5 l5: "depth" -> *"depth range"*
**A: Changed (page 5, line 12)**
p5 l5: sometimes "nitrate" is used, sometimes "NO3"
**A: We use not "nitrate" throughout the text (e.g., page 5, line 17).**
p5 l6: "ends its route down to the sea-floor" -> "stops sinking"
**A: Changed (page 5, line 13)**
p5 l8: w_D0 has already been described. *-> changed (is the derivative)*
**A: Reformulated (page 5, line 13)**
p5 l23: The reference to "ground truth" is not useful here and just confuses the reader.
**A: Changed to Genuine Truth  (page 6, line 3)**
p5 l26: "We define this model configuration as ground truth" this has now been men-
tioned multiple times
**A: Changed (page 6, line 3)**
p6 l5: "The leading thought behind these changes relative to MIX+ is to mimic the
behavior of the Genuine Truth configuration even though the vertical background dif-
fusion is substantially different to the Genuine Truth. Or, in other words, ..." This has
been mentioned before and can be simplified
**A: Reformulated (page 6, line 13)**
p6 l9: Point (1) is not part of the procedure and should appear in section 2.1 to motivate
the choice of parameters there. -> a respective motivation was added to Sect.2.1.
**A: A respective motivation was added to Sect.2.1 (page 6, line 19ff)**
p6 l14: How was chlorophyll a converted via Redfield?
**A: Sorry + we should have referred to  surface phytoplankton here such that the sentence makes more
sense + changed now (page 6, line 26).**

p6 l14: "surface chlorophyll a and oceanic phosphate": where the observations distributed evenly across the globe? Does "oceanic phosphate" imply surface values? More information is needed on how the best fit was determined.

**A:  We added the lacking information (page 6, line 27).**

p6 l19: It would be nice to actually list the bias alongside the RMSD in table 4. -> we list the biases now in the text.

**A: We added the biases to the text (e.g., page 9, line 1,2 and 7).**

p6 l30: It is not clear to me what the difference between the spinup and the drift runs are if both use pre-industrial atmospheric CO2

**A: We clarified this (page 7, line 9)**

p7 l3: This summary is repeated again in the next paragraph.

**A: Sorry, we deleted paragraph 6 and summarized everything in Section5 instead.**

p7 l15: "Mix+" -> "MIX+" -> *changed*

**A: Changed (page 8, line 1)**

p8 l1: "antagonistic": do you mean "twofold"? Please rephrase.

**A: We rephrased  this sentence (page 8, line 19).**

p8 l5 (and elsewhere): "inline" -> "in line" (or reword using for example "agree with") -> *changed*

**A: Changed**

p8 l24: "bias" but those are RMSD values, please report the bias as well.

**A: Done in Table 4 and, e.g., page 9, line 1,2,7.**

p9 l4: It would be good to mention this feedback mechanism when the model is introduced.

**A; Done (page 5, line 3).**

p9 l7: In this paragraph the language makes it seem like Genuine Truth is evaluated with respect to the TUNE configuration (same comment applies to Fig 8), it is better to switch it around, that is: "TUNE shows ... relative to Genuine Truth."

**A: Done (page 9, line 32).**

p9 l23: Remove "also" from this sentence and following introductory sentences for other subsections.

**A: Done (e.g., page 10, line 27).**

p10 l6: Doesn't "base currency" imply that variables like phytoplankton is measured in units of phosphate concentration?

**A: We agree; somehow the UVic-model output for phytoplankton was still converted to Nitrate-units and we decided to stick to this standard.**

p11 l1: Add "(b)" to figure reference (or better, switch the panel order and add "(a)").

**A: We switched the panel order of the panels in Fig. 11 as suggested.**

p12 l12: The notion of "slightly changed physics" does not seem to agree with "we have to stress that our mixing parameter settings can presumably be regarded as extreme cases" just a few lines above. I would suggest to reword, emphasizing that just one physical parameter was modified.

**A: Agreed + we rephrased the respective sentence (page 13, line 1).**

p12 l26: "The latter change is motivated...": It is difficult to follow this and the next sentence. Please rephrase or delete.

**A: We agreed + we rephrased the respective sentence (page 13, line 11ff).**

p13 l3: "is with 21 Sv huge": is this the ensemble spread from the Reintges et al. (2017) report? If so, make this explicit; if not, please rephrase, it is difficult for the reader to follow the train of thought here.  -> *we provide the actual numbers in the new version of the manuscript*

**A: We made the differences explicit (page13, line 26/27).**

p13 l8: "have been changed by 12.5, ≈200 and 115% respectively": use "%" for all numbers!

**A: Added (page 13, line 32)**

p13 l26: "In terms of the uncertainties in projected sea surface temperatures imposed by ocean mixing are locally substantial:" Something is missing in this sentence

**A: We fixed that (page 14, line 17)**

p14 l17: "suited" -> "suitable"

**A: Changed (page 15, line 5)**

Fig 2: combine with Fig 1. I would be good to show the difference between the two panels.

**A: *We added differences to both Fig1 and 2 (to be consistent).***

Fig 4, 5 and others: It would be very useful to have difference maps. If there is not enough room for extra panels I would suggest to show the regular field for Genuine Truth but difference fields for the two other configurations.

**A: We changed the Figures accordingly.**

Fig 6: A label for the y-axis is missing and the background should have a color that is not in the the color map (grey?).

**A: true + we changed the background to gray.**

Fig 6: caption says "mins"

**A: Sorry + changed that to "minus"**

Fig 7: Here, one the other hand, it may be useful to show the oxygen conc of the Genuine Truth in a new first panel.

**A: We added the respective panel.**
Fig 8: Differences in anomalies are not that easy to interpret. I suppose the temperature in 1850 is very similar in both simulations, so the difference in anomalies is basically a difference in temperature? If that is the case, mention this. Furthermore, based on the description, why is Genuine Truth - TUNE shown and not the other way around, like in all other plots? I would suggest to use three panels again: (1) anomaly for Genuine Truth, (2) difference TUNE - Genuine Truth (3) difference MIX+ - Genuine Truth. *-> we followed the suggestion of the reviewer and changed the Figure accordingly.*

Fig 11: add description and units to panel (b). It would makes sense to switch panels, so that they appear in the same order as in the manuscript text. *-> good suggestion; we changed the panels*
Table 3: combine with Table 1.
**A: Good suggestion; we combined both tables.**

[revised manuscript text omitted]

---

## Author Response (AR2)

Dear Katja Fennel,

we submitted a second revised version of the manuscript "Reciprocal bias compensation and ensuing uncertainties in model-based climate projections: pelagic biogeochemistry versus ocean mixing." by U. Löptien and H. Dietze. We found the reviewer comments again extremely helpful and included all of them into the revised version of the manuscript (you find the respective answers and line numbers of changes below). Also, we went carefully through the text once more. We hope that you find the manuscript now suitable for publication.

Thank you for your work.

Best, the authors

Answers to Reviewer #2:

*A: We are, again, very grateful to the reviewer who invested so much time to come up with very helpful and detailed suggestions. Thanks!*

R: General Comments:

R: While the structure of the manuscript has indeed improved and some statements are no longer repeated again and again, there are still a lot mistakes in the use of the English language which makes the manuscript difficult to follow in places. I have outlined some mistakes in my specific comments below but there are too many smaller mistakes to list them all. I would recommend that the authors allow a native English speaker to carefully go through the manuscript before resubmitting it.

*A: We followed this suggestion and went carefully through the text.*

R: In my first review I was curious how the parameter values for the TUNE experiment were chosen. In the revised version of the manuscript, I still cannot find the information I was looking for (it is not on page 6, line 29ff as the response to first review suggests). Furthermore, I am now wondering how the two experiments with decreased sinking fit into the picture. How were the parameter values determined, were they randomly chosen? Why are there exactly 2 simulations with decreased sinking and what are the values for the remaining parameters for those two simulations? Just a few sentences might be all that is required, at the moment the reader is not given enough information to recreate the experiment and obtain a similar reduction in model-data misfit.

*A: We concluded from the comments that mentioning the two extra simulations was rather confusing. Further, they do not add relevant content. Hence, we removed them. Additionally, the description of the remaining experiments was completed.*

Specific Comments:

R: p1 l4: "Respective model applications" -> "Applications for these models"
*A: Done (Page 1, Line 4)*

R: p1 l17: "global warming": I prefer the term "climate change" but leave this decision to the authors.
*A: True + changed (Line 17)*

R: p1 l17: "trigger ... geo-engineering options": I'd suggest "have triggered ... geo-engineering options to combat its effects on our planet."
*A: Changed (Page 1, Line 17)*

R: p2 l3: "that govern the dynamics of the set of underlying partial differential equations" -> "that are included in the model's partial differential equations which govern its dynamics"
*A: Reformulated (Page 2, Line 4)*

R: p2 l5: remove "and/" -> or
*A: Done (Page 2, Line 18)*

R: p2 l6: remove "generic" ("typical" is enough)
*A: Done (Page 2, Line 8)*

R: p2 l12: "In this study..." This sentence is not very precise, try to tie in biological models and model-data misfit reduction. Something like "In this study we illustrate how a model-data misfit arising from biases and deficiencies in the physical model can impact the estimation of biological parameters."
*A: Reformulated (Page 2, Line 14)*

R: p2 l16: "to" -> "and" -> or
*A: Done (Page 2, Line 18)*

R: p2 l19: "balancing ... away from". Do you mean "counteracting the effects of", please rephrase. -> counteracting
*A: Changed (Page 2, Line 20)*

R: p2 l27: "where actual mixing compounds explicitly prescribed mixing rates with spurious mixing ...": I had to re-read this sentence several times, please simplify and rephrase.
-> as the actual mixing results from explicitly prescribed mixing rates plus spurious mixing triggered by numerical advection and isopycnal diffusion algorithms
*A: Reformulated (Page 2, Line 30f)*

R: R: p2 l33: "we do not know the values for diapycnal diffusivity that are to be set explicitly in Earth System Models" -> "we do not have exact values for diapycnal diffusivity for use in Earth System Models" ->   the values for diapycnal diffusivity, that are to be set explicitly in Earth System Models, are poorly known
*A: Changed (Page 2, Line 35f)*

R: p2 l33: "Changes" -> "Yet, changes"
*A: Done (Page 3, Line 1)*

R: p3 l5: "Earth System Models contains several components and module after module is coupled together" -> "Earth System Models are made up of several modules which are coupled together"
*A: Reformulated (Page 3, Line 6f)*

R: p3 l6: "Thus, pelagic biogeochemistry modules are developed within a rather "fixed" physical model environment." I'd suggest "Thus, pelagic biogeochemistry modules are often developed presuming a fixed physical model component."
*A: Reformulated (Page 3, Line 8f)*

R: p3 l17: "biogeochemical mixing" This sound like a physical process, I think you mean "biogeochemical cycling".
*A: Changed (Page 3, Line 18)*

R: p3 l18: "sometimes without even the modellers knowing" Are you suggesting most modellers know about their model having two flawed model components. -> deleted (components)
*A: Deleted (Page 3, Line 19)*

R: p4 l2: "UVic" is introduced here for the second time.
*A: Deleted (Page 4, Line 1)*

R: p4 l17: "in the reference version": Is that the Genuine Truth, if so mention that.
*A: We added "Genuine Truth" as suggested (Page 4, Line 17)*

R: p4 l27: "such as differences between growth and death" -> "growth and mortality"
*A: Changed (Page 4, Line 19)*

R: p4 l32: There is a "(3)" missing here. But why not introduce these in the same order they are later presented in the equations?
*A: We added (3) and changed the order (Page 5, Line 1ff)*

R: p5 l7: "(T) up to 20C" the "up to 20C" is confusing here, I'd suggest to just use "(T in units of C)".
*A: Changed (Page 5, Line 10)*

R: p5 l14: "In one of our configurations": Name the configuration! The configurations have been introduced in the previous section and TUNE should be mentioned explicitly in this section.
*A: True + added (Page 5, Line 18)*

R: p6 l9: These 3 parameters were all just mentioned on the previous page.
*A: True + shortened (Page 6, Line 15)*

R: p6 l11: "these": the changes were not yet specified, use "the" or "our" instead.
*A: Changed to "our" (Page 6, Line 15)*

p6 l13: "changes to biogeochemical model parameters in TUNE are chosen such that the bias between the Genuine Truth and MIX+ (that is effected by the choice of a higher background diffusivity) is compensated" Be specific here, bias in what variables? If you had observations

of diffusivity, changes in the parameters could not compensate for that. However the bias in select "observed" biogeochemical variables can be reduced this way.
*A: We added the respective information (Page 6, Line 18)*

R: p6 l18: "ranges regularly" What exactly does this mean, how were the parameter values chosen?
*A: We reformulated this sentence (Page 6, Line 23/24)*

R: p6 l19: The two extra runs with decreased sinking feel a bit like an afterthought. How were the rest of the parameters chosen?
*A: We concluded from the comments that mentioning the two extra simulations was rather confusing. Further, they do not add relevant content. Hence, we removed them.*

R: p7 l27: "increases up to 1 PSU" -> "increases by up to 1 PSU"
*A: Changed (Page 8, Line 1)*

R: p8 l16: "has opposing effects" -> "has two opposing effects"
*A: Changed (Page 8, Line 21)*

R p9 l5: ". The latter" -> ", which"
*A: Reformulated + we start a new sentence now (Page 8, Line 32ff)*

R: p9 l9: Here you use "RMSE", before "RMSD" is used. -> we use now RMSE throughout the manuscript
*A: We always use RMSE now (e.g., Page 9, Line 11)*

R: p9 l20: "doubled below 1500 m depth" What exactly is doubled, the temperature difference?
*A: Reformulated (Page 8, Line 25)*

p9 l26: "are" -> "can be"
*A: Changed (Page 8, Line 30)*

R: p10 l18: Fig. 10 shows phosphate, not phytoplankton.
*A: Thanks! Changed (Page 10, Line 21)*

R: p11 l21: Here you are comparing a relative number (0.5%) to an absolute one (200GT), which makes it difficult to assess the significance of the change. Add, for example, information about the relative change to the absolute number "200Gt C (x%)". Also, change "T" to "t".
*A: Added (Page 11, Line 26)*

R: p12 l20: Here, the mixing parameter settings are described as "extreme cases", later it is argued that literature values were chosen. What is it, are the literature values extreme or are reasonable values chosen here? A bit more explanation is needed or maybe the use of "extreme" is a bit too extreme here?
*A: True - we changed "extreme" (Page 12, Line 25)*

R: p12 l29 (on p13, the line numbers keep sliding): It would be good to point out here that changes to the vertical diffusivity are the only changes to the physical model component but

that there are also changes to the biogeochemical model. If the Keller 2012 configuration only includes the physical model component, this should be pointed out in the first paragraph of Section 4.1.
*A: Thanks! We state now clearly that we here refer to the physical component of Keller 2012 (Page 13, Line 1/2)*

R: p13 l11: "To summarize: the uncertainty is substantial." I'd suggest: "To summarize: the uncertainty in the value of the vertical diffusion parameter of our physical model component is substantial and it can currently not be well constrained by either observations or experiments. Hence, ..."
*A: Reformulated (Page 13, Line 15)*

R: p13 l14: "which saves considerable computational cost": This sounds a bit like the change in advection scheme makes the model run faster, I don't think this information is needed here.
*A: we skipped the computational cost discussion since we feel that this information was not relevant in this context (Page 13, Line 18)*

R: p13 l18: Why start a new paragraph in the middle of the explanation of the effects of modifying the vertical diffusion parameter? Start new paragraph at the beginning of said explanation "<new paragraph> Using ad-hoc measures, ..."
*A: Done (Page 13, Line 19)*

R: p13 l27: It would be good to mention that the parameter values were chosen to reduce the misfit between model and data (generated by Genuine Truth).
*A: Added (Page 13, Line 32)*

R: p14 l2: "pretty" -> "quite" (or just "effective")
*A: Done (Page 14, Line 7)*

R: p14 l4: "the Genuine Truth configuration is considered state-of-the-art" I do not disagree with this assessment, however previously it is stated that "The model is relatively simple [...] and rather coarsely resolved (≈ 200km), compared to the cutting-edge generation of IPPC-type Atmosphere-Ocean General Circulation Models (AOGCMs)." You should make the argument for why this model is considered state-of-the-art in Section 4.1. Otherwise it comes as a surprise to the reader that in Section 4.3 the previously "simple" model is now considered "state-of-the-art".
*A: Agreed + changed (Page 14, Line 10/11)*

p14 l8: "As outlined above, it is hard to argue based on a-priori knowledge as concerns the differences among their underlying model parameters, which model configuration, the Genuine Truth or TUNE, is more realistic.": I repeat my argument: If one of the model configurations has "extreme" parameter values (as stated previously), then it would be reasonable to presume based on a-priori knowledge that it is less realistic.
*A: We agree with the reviewer and changed "extreme" in Set.4.1 as suggested above (Page 12, Line 25).*

R: p15 l9: To someone just reading the conclusions, you should point out that the values of the biogeochemical model parameters were chosen to counteract the effects of modifying the diffusivity.
*A: Agreed (Page 15, Line 23/24)*

R: p16 l9: "constricted" -> "constrained"
*A: Changed (Page 16, Line 14)*

R: p16 l9: "constricted" -> "constrained"
*A: Changed (Page 16, Line 14)*

[revised manuscript text omitted]

---

## Author Response (AR3)

Dear Katja Fennel,

we submitted another revised version of the manuscript "Reciprocal bias compensation and ensuing uncertainties in model-based climate projections: pelagic biogeochemistry versus ocean mixing." by U. Löptien and H. Dietze.  We included the suggested corrections as outlined below.

Thank you for your work!

Best, the authors
* * *
KF: Dear Authors,

I'm happy to accept your manuscript pending some technical corrections (see below). The first of these is a suggestion only. Thank you for supporting Open Access publications like Biogeosciences.

Best regards,
Katja Fennel

Specific Comments:

KF: In the definition of the "genuine truth" run, I don't find the word "genuine" useful or necessary. In fact, I find it misleading because this is a synthetic truth, not a genuine truth. I would suggest to either remove "genuine" throughout the manuscript (incl. in the figures) or replace "genuine" with "synthetic."
*A: We can see the point but used the same expression in an earlier paper (Löptien and Dietze, 2017, GBC) and hesitate to introduce yet another expression.*

KF: p. 3, line 10: "Here, tuning here…" – remove one "here"
*A: Thank you + changed (p3, line 10)*

KF: p. 10, line 30: Replace "phytoplankton" by "phosphate"
*A: Thank you again + changed (p10, line 30)*

KF: p. 12, line 26: "All setups are all…" – remove the 2nd all.
*A: Sorry + changed (p12, line 26)*

KF: p. 13, line 2: Replace "onliest" with "only"
*A: True + changed (p13, line 2)*

KF: p. 13, line 17: Replace "schema" with "scheme"
*A: Changed (p13, line 2)*

[revised manuscript text omitted]